# Efficiently avoiding saddle points
# with zero order methods: No gradients required

**Lampros Flokas**[*]
Department of Computer Science
Columbia University
New York, NY 10025
lamflokas@cs.columbia.edu

**Emmanouil V. Vlatakis-Gkaragkounis**[*]
Department of Computer Science
Columbia University
New York, NY 10025
emvlatakis@cs.columbia.edu

**Georgios Piliouras**
Engineering Systems and Design
Singapore University of Technology and Design
Singapore
georgios@sutd.edu.sg

## Abstract

We consider the case of derivative-free algorithms for non-convex optimization, also known as zero order algorithms, that use only function evaluations rather than gradients. For a wide variety of gradient approximators based on finite differences, we establish asymptotic convergence to second order stationary points using a carefully tailored application of the Stable Manifold Theorem. Regarding efficiency, we introduce a noisy zero-order method that converges to second order stationary points, i.e avoids saddle points. Our algorithm uses only $\tilde{\mathcal{O}}(1/\epsilon^2)$ approximate gradient calculations and, thus, it matches the converge rate guarantees of their exact gradient counterparts up to constants. In contrast to previous work, our convergence rate analysis avoids imposing additional dimension dependent slowdowns in the number of iterations required for non-convex zero order optimization.

## 1 Introduction

Given a function $f : \mathbb{R}^d \to \mathbb{R}$, solving the problem

$$\mathbf{x}^* = \underset{\mathbf{x} \in \mathbb{R}^d}{\arg \min} f(\mathbf{x})$$

is one of the building blocks that many machine learning algorithms are based on. The difficulty of this problem varies significantly depending on the properties of $f$ and the way we can access information about it. The general case of non-convex functions makes the problem significantly more challenging, since first order stationary points can be global or local optima as well as saddle points. In fact, discovering global optima is an NP hard problem in general and even for quartic functions verifying local optima is a co-NP complete problem [MK87, LPP+19].

While local optima may be satisfactory for some applications in machine learning [CHM+15], saddle points can make high dimensional non convex optimization tasks significantly more difficult [DPG+14, SQW18]. Therefore, researchers have focused their efforts on functions possessing the strict saddle property. Under this property, Hessians of $f$ evaluated at saddle points have at least one negative eigenvalue making detection of saddle points tractable. Given this assumption,

---

[*]Equal contribution

methods that use second order information like computing Hessians or Hessian-vector products [NP06, CD16, AAB$^+$17] can converge to second order stationary points (SOSPs) and thus avoid strict saddle points. Recent work [GHJY15, Lev16, JGN$^+$17, LPP$^+$19, AL18, JNJ18] has also showed that gradient descent (and its variants) can also avoid strict saddle points and converge to local minima.

Unfortunately access to gradient evaluations is not available in all settings of interest. Even with the advent of automatic differentiation software, there are several applications where computation of gradients is either computationally inefficient or even impossible. Examples of such applications are hyper-parameter tuning of machine learning models [SLA12, SHCS17, CRS$^+$18], black-box adversarial attacks on deep neural networks [PMG$^+$17, MMS$^+$18, CZS$^+$17], computer network control [LCCH18], variational approaches to graphical models [WJ08] and simulation based [RK16, Spa03] or bandit feedback optimization [ADX10, CG19]. Zero order methods, also known as black-box methods, try to address these issues by employing only evaluations of the function $f$ during the optimization procedure. The case of convex functions is well understood [NS17, DJWW15, ADX10]. For the non-convex case, there has been a considerable amount of work on the convergence to first order stationary points both for deterministic settings [NS17] and stochastic ones [GL13, WDBS18, BG18, LKC$^+$18, GHH16].

The case of SOSPs has been so far comparatively under-studied. It has been established that SOSPs are achievable through zero order trust region methods that employ fully quadratic models [CSV09]. The disadvantage of trust region methods is that their computation cost per iteration is $\mathcal{O}(d^4)$ which becomes quickly prohibitive as we increase the number of dimensions $d$. More recently, the authors of [JLGJ18] studied the case of finding local minima of functions having access only to approximate function or gradient evaluations. They manage to reduce zero order optimization to the stochastic first order optimization of a Gaussian smoothed version of $f$. While this approach yields guarantees of convergence to SOSPs , each stochastic gradient evaluation requires $\mathcal{O}(\text{poly}(d, 1/\epsilon))$ number of function evaluations. This leads to significantly less efficient optimization algorithms when compared to their first order counterparts. **It is therefore yet unclear if there are scalable zero order methods that can safely avoid strict saddle points and always converge to local minima of $f$. To the best of our knowledge, our work is the first one to establish a positive answer to this important question.**

**Our results.** *We prove that zero order optimization methods solve general non-convex problems efficiently.* In a nutshell, we present a family of of zero order optimization methods which provably converge to SOSPs . Our proof includes a new, elaborating analysis of Stable Manifold Theorem (See Section 4). Additionally, the number of the approximate gradient evaluations match the standard bounds for first order methods in non-convex problems (see Table 1 & Section 5).

| Algorithm | Oracle | Iterations | Evaluations of $f$ |
|---|---|---|---|
| **Theorem** 3 | Approximate Gradient | Asymptotic | Asymptotic |
| [LPP$^+$19] | Exact Gradient | Asymptotic | - |
| **Theorem** 4 | Approx. Gradient + Noise | $\tilde{\mathcal{O}}(1/\epsilon^2)$ | $\tilde{\mathcal{O}}(d/\epsilon^2)$ |
| FPSGD [JLGJ18] | Approx. Gradient + Noise | $\tilde{\mathcal{O}}(d/\epsilon^2)$ | $\tilde{\mathcal{O}}(d^4/\epsilon^4)$ |
| ZPSGD [JLGJ18] | Function Evaluations + Noise | $\tilde{\mathcal{O}}(1/\epsilon^2)$ | $\tilde{\mathcal{O}}(d^2/\epsilon^5)$ |
| [JGN$^+$17] | Exact Gradient + Noise | $\tilde{\mathcal{O}}(1/\epsilon^2)$ | - |

Table 1: Oracle model and iteration complexity to SOSPs .

**Algorithms.** Instead of focusing on a single finite differences algorithms, we construct a general framework of approximate gradient oracles that generalizes over many finite differences approaches in the literature. We then use these approximate gradient oracles to devise approximate gradient descent algorithms. For more details see Section 3.3 and Definition 4.

**Asymptotic convergence.** We use the stable manifold theorem to prove that zero order methods can almost surely avoid saddle points. In contrast to the analysis of [LPP$^+$19] for first order methods, the zero order case is more demanding. Convergence to first order stationary points requires changing the gradient approximation accuracy over the iterations and, thus, the equivalent dynamical system is time dependent. By reducing our time dependent dynamical system to a time invariant one defined in

an expanded state, we are able to obtain provable guarantees about avoiding saddle points. To extend our guarantees of convergence to deterministic choices of the initial accuracy, we provide a carefully tailored application of the Stable Manifold Theorem that analyzes the structure of the stable manifolds of the dynamical system. Our results on saddle point avoidance extend to functions with non isolated critical points. To address this, we provide sufficient conditions for point-wise convergence of the iterates of approximate gradient descent methods for the case of analytic functions.

**Convergence rates for noisy dynamics.** In order to produce fast convergence rates, as in the case of first order methods [JGN$^+$17], it is useful to consider perturbed/noisy versions of the dynamics. Once again the case of zero order methods poses distinct hurdles. Close to critical points of $f$, approximations of the potentially arbitrarily small gradient can be very noisy. Iterates of exact gradient descent and approximate gradient descent may diverge significantly in this case. In fact, provably escaping saddle points by guaranteeing decrease of value of $f$ is more challenging for the case of approximate gradient descent since it is not a descent algorithm. A key technical step is to show that the negative curvature dynamics that enable gradient descent to escape saddle points are robust to gradient approximation errors. As long as the gradient approximation error is smaller than a fixed a-priori known threshold, zero order methods can provably escape saddle points. Based on this, we are able to prove that zero order methods can converge to approximate SOSPs with the same number of approximate gradient evaluations provided by [JGN$^+$17] up to constants.

It is worth pointing out that achieving an $\tilde{\mathcal{O}}(\epsilon^{-2})$ bound of approximate gradient evaluations requires conceptually different techniques from other recent approaches in zero order methods. Indeed, previous work on randomized and stochastic zero order optimization [NS17, GL13] has relied on treating randomized approximate gradients of $f$ as in expectation exact gradients of a carefully constructed smoothed version of $f$. Then with some additional work, convergence arguments for the smooth version of $f$ can be transferred to $f$ itself. Although these arguments are applicable to our case as well, as shown by the work of [JLGJ18], they also lead to a slowdown both in terms of the dimension $d$ and the required accuracy $\epsilon$. The main reasons behind this slowdown are that the Lipschitz constants of the smoothed version of $f$ depend on $d$ and the high variance of the stochastic gradient estimators. To sidestep both issues, we analyze the effect of gradient approximation error directly on the optimization of $f$.

## 2 Related Work

Our work builds and improves upon previous finite difference approaches for non-convex optimization and provides SOSP guarantees previously only reserved to computationally expensive methods.

**First Order Algorithms** A recent line of work has shown that gradient descent and variations of it can actually converge to SOSPs . Specifically, [LPP$^+$19] shows that gradient descent starting from a random point can eventually converge to SOSPs with probability one. [JGN$^+$17, JNJ18] modified standard gradient descent using perturbations to provide an algorithm that converges to SOSPs in $\mathcal{O}(\text{poly}(\log d, 1/\epsilon))$ iterations. As noted in the introduction, the zero order case poses additional hurdles compared to the first order one. Our work, by addressing these hurdles effectively extends the guarantees provided by [LPP$^+$19, JGN$^+$17] to zero order methods.

**Zero Order Algorithms** Approximating gradients using finite differences methods has been the standard approach for both for convex and non-convex zero order optimization.[NS17] established convergence properties even for randomized gradient oracles. Recently, [DJWW15] provided optimal guarantees for stochastic convex optimization up to logarithmic factors. For the more general case of stochastic non-convex optimization there has been extensive work covering several aspects of the problem: distributed [HZ18], asynchronous [LZH$^+$16], high-dimensional [WDBS18, BG18] optimization and variance reduction [LKC$^+$18, GHH16]. It is significant to mention that the aforementioned work is focused on convergence to $\epsilon-$first order stationary points.

Regarding SOSPs , [CSV09] showed that trust region methods that employ fully quadratic models can converge to SOSPs at the cost of $\mathcal{O}(d^4)$ operations per iteration. The authors of [JLGJ18] studied the convergence to SOSPs using approximate function or gradient evaluations. While both approaches are applicable for the zero order setting with exact function evaluations, as we will see in Section 3.4, this type of reduction results in algorithms that require substantially more function evaluations to reach an $\epsilon$-SOSP . Our work provides provable guarantees of convergence at significantly faster rates.

# 3 Preliminaries

## 3.1 Notation

We will use lower case bold letters $\mathbf{x}, \mathbf{y}$ to denote vectors. $\|\cdot\|$ will be used to denote the spectral norm and the $\ell_2$ vector norm. $\lambda_{min}(\cdot)$ will be used to denote the minimum eigenvalue of a matrix. If $g$ is a vector valued differentiable function then $Dg$ denotes the differential of function $g$. We will use $\{e_1, e_2, \ldots e_d\}$ to refer to the standard orthonormal basis of $\mathbb{R}^d$. Also $C^n$ is the set of $n$ times continuously differentiable functions. $B_{\mathbf{x}}(r)$ refers to the ball of radius $r$ centered at $\mathbf{x}$. Finally, $\mu(S)$ is the Lebesgue measure of a measurable set $S \subseteq \mathbb{R}^d$.

## 3.2 Definitions

A function $f : \mathbb{R}^d \to \mathbb{R}$ is said to be $L$-continuous, $\ell$-gradient, $\rho$-Hessian Lipschitz if for every $\mathbf{x}, \mathbf{y} \in \mathbb{R}^d \|f(\mathbf{x}) - f(\mathbf{y})\| \leq L\|\mathbf{x} - \mathbf{y}\|, \|\nabla f(\mathbf{x}) - \nabla f(\mathbf{y})\| \leq \ell\|\mathbf{x} - \mathbf{y}\|, \|\nabla^2 f(\mathbf{x}) - \nabla^2 f(\mathbf{y})\| \leq \rho\|\mathbf{x} - \mathbf{y}\|$ correspondingly. Additionally, we can define approximate first order stationary points as:

**Definition 1** ($\epsilon$-first order stationary point). *Let $f : \mathbb{R}^d \to \mathbb{R}$ be a differentiable function. Then $\mathbf{x} \in \mathbb{R}^d$ is a first order stationary point of $f$ if $\|\nabla f(\mathbf{x})\| \leq \epsilon$.*

A first order stationary point can be either a local minimum, a local maximum or a saddle point. Following the terminology of [LPP+19] and [JGN+17], we will include local maxima in saddle points since they are both undesirable for our minimization task. Under this definition, strict saddle points can be identified as follows:

**Definition 2** (Strict saddle point). *Let $f : \mathbb{R}^d \to \mathbb{R}$ be a twice differentiable function. Then $\mathbf{x} \in \mathbb{R}^d$ is a strict saddle point of $f$ if $\|\nabla f(\mathbf{x})\| = 0$ and $\lambda_{min}(\nabla^2 f(\mathbf{x})) < 0$.*

To avoid convergence to strict saddle points, we need to converge to SOSPs . In order to study the convergence rate of algorithms that converge to SOSPs , we need to define some notion of approximate SOSPs . Following the convention of [JGN+17] we define the following:

**Definition 3** ($\epsilon$-SOSP ). *Let $f : \mathbb{R}^d \to \mathbb{R}$ be a $\rho$-Hessian Lipschitz function. Then $\mathbf{x} \in \mathbb{R}^d$ is an $\epsilon$-second order order stationary point of $f$ if $\|\nabla f(\mathbf{x})\| \leq \epsilon$ and $\lambda_{min}(\nabla^2 f(\mathbf{x})) \geq -\sqrt{\rho\epsilon}$.*

## 3.3 Gradient Approximation using Zero Order Information

One of the key ways that enables zero order methods to converge quickly is using approximations of the gradient based on finite differences approaches. Here we will show how forward differencing can provide these approximate gradient calculations. Without much additional effort we can get the same results for other finite differences approaches like backward and symmetric difference as well as finite differences approaches with higher order accuracy guarantees. Let us define the gradient approximation function based on forward difference $r_f : \mathbb{R}^d \times \mathbb{R} \to \mathbb{R}^d$

$$r_f(\mathbf{x}, h) = \begin{cases} \sum_{l=0}^{d} \dfrac{f(\mathbf{x} + h\mathbf{e}_l) - f(\mathbf{x})}{h} \mathbf{e}_l \text{ when } h \neq 0 \\ \nabla f(\mathbf{x}) \text{ if } h = 0 \end{cases} \tag{1}$$

This function takes two arguments: A vector $\mathbf{x}$ where the gradient should be approximated as well as a scalar value $h$ that controls the approximation accuracy of the estimator. An additional property that will be of interest when we analyze approximate gradient descent is the fact that $r_f$ is Lipschitz. Based on the definition one can show:

**Lemma 1.** *Let $f$ be $\ell$-gradient Lipschitz. Then $r_f(\cdot, h)$ as defined in Equation 1 is $\sqrt{d}\ell$ Lipschitz for all $h \in \mathbb{R}$ and $\forall h \in \mathbb{R}, \mathbf{x} \in \mathbb{R}^d : \|r_f(\mathbf{x}, h) - \nabla f(\mathbf{x})\| \leq \ell\sqrt{d}|h|$.*

## 3.4 Black box reductions to first order methods

As shown in the works of [NS17, GL13], zero order optimization is reducible to stochastic first order optimization. The reduction relies on treating randomized approximate gradients of $f$ as in expectation exact gradients of a carefully constructed smoothed version of $f$. These arguments are also applicable to our case as well. FPSG, one of the approaches of [JLGJ18], naively leads to a large

poly($d$) dependence in the convergence rate. More specifically one can show that [JLGJ18]'s FPSG method needs $\tilde{\mathcal{O}}(d^3/\epsilon^4)$ evaluations of $\nabla g$ to converge to an $\epsilon$-SOSP . The main reason behind this dimension dependent slowdown is that the Hessian Lipschitz constant of the smoothed version of $g$ is $O(\rho\sqrt{d})$. An alternative approach in [JLGJ18] named ZPSG builds gradient estimators using function evaluations directly. The main source slowdown here is the high variance of the stohastic gradients. An analysis of those methods for the case where exact function evaluations are available can be found in the Appendix.

In the next sections we will provide an alternative analysis that accounts for the gradient approximation errors on the optimization of $f$ directly. Thus, we will be able to sidestep the above issues and provide faster convergence rates and better sample complexity.

# 4 Approximate Gradient Descent

## 4.1 Description

It is easy to see that conceptually any iterative optimization method can be expressed as a dynamical system of the form $\{\mathbf{x}_{k+1} = g(\mathbf{x}_k)\}$ where $\mathbf{x}_k$ is the current solution iterate that gets updated through an update function $g$. Additionally, for first order methods strict saddle points correspond to the unstable fixed points of the dynamical system. These key observations have motivated [LPP$^+$19] to use the Stable Manifold Theorem (SMT) [Shu87] in order to prove that gradient descent avoids strict saddle points. Intuitively, SMT formalizes why convergence to unstable fixed points is unlikely starting from a local region around an unstable fixed point. Adding the requirement that $g$ is a global diffeomorphism, [LPP$^+$19] generalizes the conclusions of SMT to the whole space.

In order to prove similar guarantees for a zero order algorithm using approximate gradient evaluations, we will need to construct a new dynamical system that is applicable to our zero order setting. The state of our dynamical system $\boldsymbol{\chi}_k$ consists of two parts: The current solution iterate $\mathbf{x}_k$ that is a vector in $\mathbb{R}^d$ and a scalar value $h \in \mathbb{R}$ that controls the quality of the gradient approximation. Specifically we have

$$\boldsymbol{\chi}_{k+1} = g_0(\boldsymbol{\chi}_k) \triangleq \begin{pmatrix} \mathbf{x}_{k+1} \\ h_{k+1} \end{pmatrix} = \begin{pmatrix} \mathbf{x}_k - \eta q_x(\mathbf{x}_k, h_k) \\ \beta q_h(h_k) \end{pmatrix} \tag{2}$$

where $\eta, \beta \in \mathbb{R}^+$ positive scalar parameters and functions $q_x : \mathbb{R}^d \times \mathbb{R} \to \mathbb{R}^d$ and $q_h : \mathbb{R} \to \mathbb{R}$. The function $q_x$ can be seen as the gradient approximation oracle used by the dynamical system as described in Section 3.3. The function $q_h$ is responsible for controlling the accuracy of the gradient approximation. As we shall see later, it is important that $h_k$ converges to 0 so that the stable points of $g_0$ are the same as in gradient descent.

## 4.2 Avoiding Strict Saddle points

In this section we will provide sufficient conditions that the parameters $\eta, \beta$ must satisfy so that the update rule of Equation 2 avoids convergence to strict saddle points. To do this we will need to introduce some properties of $g_0$.

**Definition 4** (($L, B, c$)-Well-behaved function). *Let $f : \mathbb{R}^d \to \mathbb{R} \in C^2$ be a $\ell$-gradient Lipschitz function. A function $g_0$ of the form of Equation 2 is a $(L, B, c)$-well behaved function (for function $f$) if it has the following properties: i) $q_x, q_h \in C^1$ with $q_h(0) = 0$. ii) $\forall h \in \mathbb{R} : q_x(\cdot, h)$ is $L$ Lipschitz and $0 < \frac{\partial q_h(h)}{\partial h} \leq B$. iii) $\forall (\mathbf{x}, h) \in \mathbb{R}^{d+1} : \|q_x(\mathbf{x}, h) - \nabla f(\mathbf{x})\| \leq c|h|$.*

Given this definition and Lemma 1, it is clear that we can always construct $(L, B, c)$-well-behaved functions for $L = \sqrt{d}\ell$, $B = 1$, $c = \sqrt{d}\ell$ using $q_x = r_f$ and $q_h = h$.

In the following lemmas and theorems we will require that $\beta B < 1$. Under this assumption $\beta q_h$ is a contraction having 0 as its only fixed point so for all fixed points of $g_0$ we know that $h = 0$. Notice also that when $h = 0$, we have $q_x(\mathbf{x}, 0) = \nabla f(\mathbf{x})$ and therefore the $\mathbf{x}$ coordinates of fixed points of $g_0$ must coincide with first order stationary points of $f$. In fact, in the Appendix we prove that there is a one to one mapping between strict saddles of $f$ and unstable fixed points of $g_0$. Using the same assumptions, we also get that $\det(\mathrm{D}g_0(\cdot)) \neq 0$. Putting all together, we are able to prove our first main result.

**Theorem 1.** *Let $g_0$ be a $(L, B, c)$-well-behaved function for function $f$. Let $X_f^*$ be the set of strict saddle points of $f$. Then if $\eta < \frac{1}{L}$ and $\beta < \frac{1}{B}$: $\forall h_0 \in \mathbb{R} : \mu(\{\mathbf{x}_0 : \lim_{k \to \infty} \mathbf{x}_k \in X_f^*\}) = 0$.*

Notice that the random initialization refers only to the $\mathbf{x}_0$'s domain. Indeed a straightforward application of the result of [LPP+19] would guarantee a saddle-avoidance lemma only under an extra random choice of $h_0$. Such a result would not be able to clarify if saddle-avoidance stems from the instability of the fixed point, just like in first order methods, or from the additional randomness of $h_0$. The key insight provided by the SMT is that the all the initialization points that eventually converge to an unstable fixed point lie in a low dimensional manifold. Thus, to obtain a stronger result we have to understand how SMT restricts the dimensionality of this stable manifold for a fixed $h_0$. The structure of the eigenvectors of the Jacobian of $g_0$ around a fixed point reveals that such an interesting decoupling is finally achievable.

### 4.3 Convergence

In the previous section we provided sufficient conditions to avoid convergence to strict saddle points. These results are meaningful however only if $\lim_{k \to \infty} \mathbf{x}_k$ exists. Therefore, in this section we will provide sufficient conditions such that the dynamic system of $g_0$ converges. Given that strict saddle points are avoided, it is sufficient to prove convergence to first order stationary points. Let the error of the gradient approximation be $\varepsilon_k = q_x(\mathbf{x}_k, h_k) - \nabla f(\mathbf{x}_k)$. Firstly we establish the zero order analogue of the folklore lower bound for the decrease of the function:

**Lemma 2** (Step-Convergence). *Suppose that $g_0$ is a $(L, B, c)$-well-behaved function for a $\ell$-gradient Lipschitz function $f$. If $\eta \leq \frac{1}{\ell}$ then we have that $f(\mathbf{x}_{k+1}) \leq f(\mathbf{x}_k) - \frac{\eta}{2}\left(\|\nabla f(\mathbf{x}_k)\|^2 - \|\varepsilon_k\|^2\right)$.*

Given this lemma we can prove convergence to first order stationary points.

**Theorem 2** (Convergence to first order stationary points). *Suppose that $g_0$ is a $(L, B, c)$-well-behaved function for a $\ell$-gradient Lipschitz function $f$. Let $\eta \leq \frac{1}{\ell}$, $\beta < \frac{1}{B}$. Then if $f$ is lower bounded $\lim_{k \to \infty} \|\nabla f(\mathbf{x}_k)\| = 0$.*

The last theorem gives us a guarantee that the norm of the gradient is converging to zero but this is not enough to prove convergence to a single stationary point if $f$ has non isolated critical points. In the Appendix, we prove that if the gradient approximation error decreases quickly enough then convergence to a single stationary point is guaranteed for analytic functions. This allows us to conclude our analysis with this final theorem.

**Theorem 3** (Convergence to minimizers). *Let $f : \mathbb{R}^d \to \mathbb{R} \in C^2$ be a $\ell$-gradient Lipschitz function. Let us also assume that $f$ is analytic, has compact sub-level sets and all of its saddle points are strict. Let $g_0$ be a $(L, B, c)$-well-behaved function for $f$ with $\eta < \min\{\frac{1}{L}, \frac{1}{2\ell}\}$ and $\beta < \frac{1-2\eta\ell}{B}$. If we pick a random initialization point $\mathbf{x}_0$, then we have that for the $\mathbf{x}_k$ iterates of $g_0$*

$$\forall h_0 \in \mathbb{R} : \quad \Pr(\lim_{k \to \infty} \mathbf{x}_k = \mathbf{x}^*) = 1$$

*where $\mathbf{x}^*$ is a local minimizer of $f$.*

## 5 Escaping Saddle Points Efficiently

### 5.1 Overview

In the previous subsections we provided sufficient conditions for approximate gradient descent to avoid strict saddle points. However, the stable manifold theorem guarantees that this will happen asymptotically. In fact, convergence could be quite slow until we reach a neighborhood of a local minimum. An analysis done for the first order case by [DJL+17] showed that avoiding saddle points could take exponential time in the worst case. In this section, we will use ideas from the work of [JGN+17] in order to get a zero order algorithm that converges to SOSPs efficiently.

Convergence to SOSPs poses unique challenges to zero order methods when it comes to controlling the gradient approximation accuracy. For convergence to first order stationary points one can use property iii) of Definition 4 and Lemma 2 to show that $h = \epsilon/c$ guarantees the decrease of $f$ until $\|\nabla f(\mathbf{x}_k)\| \leq \epsilon$. For SOSPs , this is not applicable as the norm of the gradient can become arbitrarily small near saddle points. One could resort to iteratively trying smaller $h$ to find one that guarantees

the decrease of $f$. A surprising fact about our algorithm is that even if the gradient is arbitrarily small, computationally burdensome searches for $h$ can be totally avoided.

## 5.2 Algorithm

**Algorithm** Initialization: $(\ell, \rho, \epsilon, c, \delta, \Delta_f)$

1: $\chi \leftarrow 3 \max\{\log(\frac{d\ell\Delta_f}{c\epsilon^2\delta}), 4\}$, $\eta \leftarrow \frac{c}{\ell}$, $r \leftarrow \frac{\sqrt{c}}{\chi^2} \cdot \frac{\epsilon}{\ell}$, $g_{\text{thres}} \leftarrow \frac{\sqrt{c}}{\chi^2} \cdot \epsilon$, $f_{\text{thres}} \leftarrow \frac{c}{\chi^3} \cdot \sqrt{\frac{\epsilon^3}{\rho}}$

2: $t_{\text{thres}} \leftarrow \frac{\chi}{c^2} \cdot \frac{\ell}{\sqrt{\rho\epsilon}}$, $S \leftarrow \frac{\sqrt{c}}{\chi} \frac{\sqrt{\rho\epsilon}}{\rho}$, $h_{low} \leftarrow \frac{1}{c_h} \min\{g_{\text{thres}}, \frac{r\rho\delta S}{2\sqrt{d}}\}$

| **Algorithm 1** PAGD($\mathbf{x}_0$) | **Algorithm 2** EscapeSaddle ($\hat{\mathbf{x}}$) |
|---|---|
| 1: **for** $t = 0, 1, \ldots$ **do** | 1: $\boldsymbol{\xi} \sim \text{Unif}(B_{\mathbf{0}}(r))$ |
| 2: $\quad$ $\mathbf{z}_t \leftarrow q(\mathbf{x}_t, \frac{g_{\text{thres}}}{4c_h})$ | 2: $\tilde{\mathbf{x}}_0 \leftarrow \hat{\mathbf{x}} + \boldsymbol{\xi}$ |
| 3: $\quad$ **if** $\|\mathbf{z}_t\| \geq \frac{3}{4} g_{\text{thres}}$ **then** | 3: **for** $i = 0, 1, \ldots t_{\text{thres}}$ **do** |
| 4: $\quad\quad$ $\mathbf{x}_{t+1} \leftarrow \mathbf{x}_t - \eta\mathbf{z}_t$ | 4: $\quad$ **if** $f(\hat{\mathbf{x}}) - f(\tilde{\mathbf{x}}_i) \geq f_{\text{thres}}$ **then** |
| 5: $\quad$ **else** | 5: $\quad\quad$ **return** $\tilde{\mathbf{x}}_i$ |
| 6: $\quad\quad$ $\mathbf{x}_{t+1} \leftarrow$ EscapeSaddle ($\mathbf{x}_t$) | 6: $\quad$ **end if** |
| 7: $\quad\quad$ **if** $\mathbf{x}_{t+1} = \mathbf{x}_t$ **then return** $\mathbf{x}_t$ | 7: $\quad$ $\tilde{\mathbf{x}}_{i+1} \leftarrow \tilde{\mathbf{x}}_i - \eta q(\tilde{\mathbf{x}}_i, h_{low})$ |
| 8: $\quad$ **end if** | 8: **end for** |
| 9: **end for** | 9: **return** $\hat{\mathbf{x}}$ |

Just like [JGN+17], we will assume that $f$ is $\ell$−gradient Lipschitz and also $\rho$−Hessian Lipschitz. To construct a zero order algorithm we will also need a gradient approximator $q : \mathbb{R}^d \times \mathbb{R} \to \mathbb{R}^d$. We will only require the error bound property on $q$, i.e., there exists a constant $c_h$ such that

$$\forall \mathbf{x} \in \mathbb{R}^d, h \in \mathbb{R} : \|q(\mathbf{x}, h) - \nabla f(\mathbf{x})\| \leq c_h |h|$$

The high level idea of Algorithm 1 is that given a point $\mathbf{x}_t$ that is not an $\epsilon$-SOSP the algorithm makes progress by finding a $\mathbf{x}_{t+1}$ where $f(\mathbf{x}_{t+1})$ is substantially smaller than $f(\mathbf{x}_t)$. By the definition of $\epsilon$-SOSPs either the gradient of $f$ at $\mathbf{x}_t$ is large or the Hessian has a substantially negative eigenvalue.

Separating these two cases is not as straightforward as in the first order case. Given the norm of the approximate gradient $q(\mathbf{x}, h)$, we only know that $\|\nabla f(\mathbf{x})\| \in \|q(\mathbf{x}, h)\| \pm c_h|h|$. In Algorithm 1 by choosing $3g_{\text{thres}}/4$ as the threshold to test for and $h = g_{\text{thres}}/(4c_h)$, we guarantee that in step 4 $\|\nabla f(\mathbf{x}_t)\| \geq g_{\text{thres}}/2$. This threshold is actually high enough to guarantee substantial decrease of $f$. Indeed given that we have a lower bound on the exact gradient and using Lemma 2 we get

$$f(\mathbf{x}_t) - f(\mathbf{x}_{t+1}) \geq \frac{\eta}{2} \left( \|\nabla f(\mathbf{x}_t)\|^2 - \|\boldsymbol{\varepsilon}_t\|^2 \right) \geq \tfrac{3}{32}\eta g_{\text{thres}}^2$$

where $\boldsymbol{\varepsilon}_t$ is the gradient approximation error at $\mathbf{x}_t$. This decrease is the same as in the first order case up to constants.

On the other hand, in Algorithm 2 we are guaranteed that $\|\nabla f(\hat{\mathbf{x}})\| \leq g_{\text{thres}}$. In this case our approximate gradient cannot guarantee a substantial decrease of $f$. However, we know that the Hessian has a substantially negative eigenvalue and therefore a direction of steep decrease of $f$ must exist. The problem is that we do not know which direction has this property. In [JGN+17] it is proved that identifying this direction is not necessary for the first order case. Adding a small random perturbation to our current iterate (step 2) is enough so that with high probability we can get a substantial decrease of $f$ after at most $t_{\text{thres}}$ gradient descent steps (step 5). Of course this work is not directly applicable to our case since we do not have access to exact gradients.

The work of [JGN+17] mainly depends on two arguments to provide its guarantees. The first argument is that if the $\tilde{\mathbf{x}}_i$ iterates do not achieve a decrease of $f_{\text{thres}}$ in $t_{\text{thres}}$ steps then they must remain confined in a small ball around $\tilde{\mathbf{x}}_0$. Specifically for the exact gradient case we have that

$$\|\tilde{\mathbf{x}}_i - \tilde{\mathbf{x}}_0\|^2 \leq 2\eta f_{\text{thres}} t_{\text{thres}}.$$

The zero order case is definitely more challenging since each update in Algorithm 2 is not guaranteed to decrease the value of $f$. Therefore, iterates may wander away from $\tilde{\mathbf{x}}_0$ without even decreasing the function value of $f$. To amend this argument for the zero order case we require that $h_{low} \leq g_{\text{thres}}/c_h$. This guarantees that even if gradient approximation errors amass over the iterations we will get the same bound as the first order case up to constants.

The second argument of [JGN$^+$17] formalizes why the existence of a negative eigenvalue of the Hessian is important. Let us run gradient descent starting from two points $\mathbf{u}_0$ and $\mathbf{w}_0$ such that $\mathbf{w}_0 - \mathbf{u}_0 = \kappa \mathbf{e}$ where $\mathbf{e}$ is the eigenvector corresponding to the most negative eigenvalue of the Hessian and $\kappa \geq r\delta/(2\sqrt{d})$. Then at least one of the sequences $\{\mathbf{w}_i\}, \{\mathbf{u}_i\}$ is able to escape away from its starting point in $t_{\text{thres}}$ iterations and by the first argument it is also able to decrease the value of $f$ substantially. The proof of the claim is based on creating a recurrence relationship on $\mathbf{v}_i = \mathbf{w}_i - \mathbf{u}_i$. The corresponding recurrence relationship for the zero order case is more complicated with additional terms that correspond to the gradient approximation errors for $\mathbf{w}_i$ and $\mathbf{u}_i$. However, we are able to prove that if $h_{low} \leq r\rho\delta S/(2\sqrt{d})$ then these additional terms cannot distort the exponential growth of $\mathbf{v}_i$. Having extended both arguments of [JGN$^+$17] we can establish the same guarantees for escaping saddle points.

**Theorem 4** (Analysis of PAGD). *There exists absolute constant $c_{\max}$ such that: if $f$ is $\ell$-gradient Lipschitz and $\rho$-Hessian Lipschitz, then for any $\delta > 0, \epsilon \leq \frac{\ell^2}{\rho}, \Delta_f \geq f(\mathbf{x}_0) - f^\star$, and constant $c \leq c_{\max}$, with probability $1 - \delta$, the output of PAGD$(\mathbf{x}_0, \ell, \rho, \epsilon, c, \delta, \Delta_f)$ will be an $\epsilon$-SOSP , and have the following number of iterations until termination:*

$$\mathcal{O}\left( \frac{\ell(f(\mathbf{x}_0) - f^\star)}{\epsilon^2} \log^4\left( \frac{d\ell\Delta_f}{\epsilon^2\delta} \right) \right)$$

## 6   Experiments

In this section we use simulations to verify our theoretical findings. Specifically we are interested in verifying if zero order methods can avoid saddle points as efficiently as first order methods. To do this we use the two dimensional Rastrigin function, a popular benchmark in the non-convex optimization literature. This function exhibits several strict saddle points so it will be an adequate benchmark for our case. The two dimensional Rastrigin function can be defined as

$$\text{Ras}(x_1, x_2) = 20 + x_1^2 - 10\cos(2\pi x_1) + x_2^2 - 10\cos(2\pi x_2).$$

For this experiment we selected 75 points randomly from $[-1.5, 1.5] \times [-1, 5, 1.5]$. In this domain the Rastrigin function is $\ell$-gradient Lipschitz with $\ell \approx 63.33$. Using these points as initialization we run gradient descent and the approximate gradient descent dynamical system we introduced in Section 4.2. For both gradient descent and approximate gradient descent we used $\eta = 1/(4\ell)$. Then for approximate gradient descent we used symmetric differences to approximate the gradients and $\beta = 0.95$ as well as $h_0 = 0.15$. Figure 1 shows the contour plot of the Rastrigin function as well

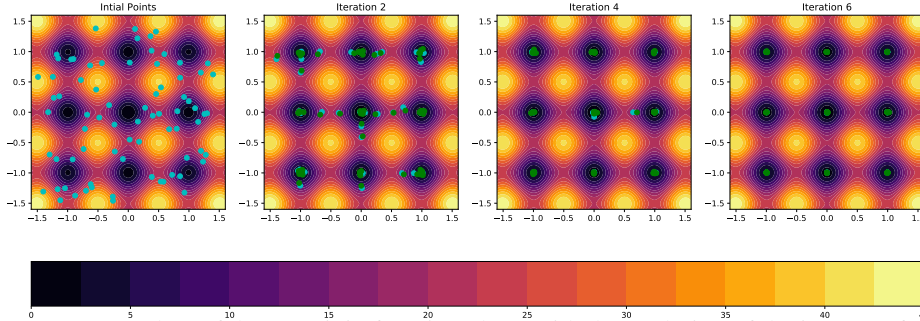

Figure 1: Contour plots of the Rastrigin function along with the evolution of the iterates of gradient descent and approximate gradient descent. Green points correspond to gradient descent whereas cyan points correspond to approximate gradient descent.

as the evolution of the iterates of both methods. As expected, for points initialized closed to local minima of the function convergence is quite fast. On the other hand, points starting close to saddle points of the Rastrigin function take some more time to converge to minima. However, it is clear that in both cases the behaviour of gradient descent and approximate gradient descent is similar in the sense that for the same initialization there is no discrepancy in terms of convergence speed for the two methods.

We also want to experimentally verify the performance of PAGD. To do this we use the octopus function proposed by [DJL$^+$17]. This function is is particularly relevant to our setting as it possesses

a sequence of saddle points. The authors of [DJL$^+$17] proved that for this function gradient descent needs exponential time to avoid saddle points before converging to a local minimum. In contrast the perturbed version of gradient descent (PGD) of [JGN$^+$17] does not suffer from the same limitation. Based on the results of Theorem 4, we expect PAGD to not have this limitation as well. We compare gradient descent (GD), PGD, AGD and PAGD on an octopus function of $d = 15$ dimensions. Figure 2 clearly shows that the zero order versions have the same iteration performance with the first-order ones. In fact, AGD is shown to behave even better than GD in this example thanks to the noise induced by the gradient approximation.

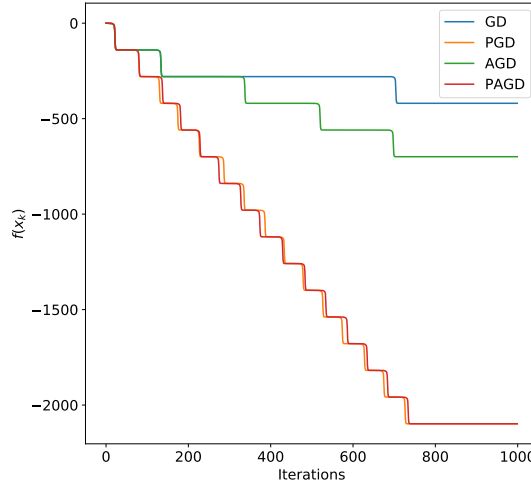

Figure 2: Octopus function value varying the number of iterations. Parameters of the function $\tau = e$, $L = e$, $\gamma = 1$. Parameters of first order methods taken from [DJL$^+$17]. Zero order methods use symmetric differencing with $h = 0.01$

# 7 Conclusion

This paper is the first one to establish that zero order methods can avoid saddle points efficiently. To achieve this we went beyond smoothing arguments used in prior work and studied the effect of the gradient approximation error on first order methods that converge to second order stationary points. One important open question for future work is whether similar guarantees can be established for other zero order methods used in practice like direct search methods and trust region methods using linear models. Another generalization of interest would be to consider the performance of zero order methods for instances of (non-convex) constrained optimization.

# Acknowledgements

Georgios Piliouras acknowledges MOE AcRF Tier 2 Grant 2016-T2-1-170, grant PIE-SGP-AI-2018-01 and NRF 2018 Fellowship NRF-NRFF2018-07. Emmanouil-Vasileios Vlatakis-Gkaragkounis was supported by NSF CCF-1563155, NSF CCF-1814873, NSF CCF-1703925, NSF CCF-1763970. We are grateful to Alexandros Potamianos for bringing this problem to our attention, and for helpful discussions at an early stage of this project for its connection to Natural Language Processing tasks. Finally, this work was supported by the Onassis Foundation - Scholarship ID: F ZN 010-1/2017-2018.

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
