[Supplementary Material]

# Efficiently avoiding saddle points
# with zero order methods: No gradients required
## Supplementary Materials

## A Preliminaries Detailed proofs

In this first subsection, we show that the forward finite differences method can be used to construct an approximate gradient oracle. Similar oracles can be constructed using backward, symmetric finite differences or Richardson extrapolation which have even higher gradient approximation accuracy. Additionally, we compute the Lipschitz constant of our method and we show that our definition of "well-behaved" approximate gradient is well defined. In other words, there are simple approximation oracles which follow the smoothness requirements that our work assumes.

### A.1 Gradient Approximation using Zero Order Information

**Lemma 4** ( Lemma 1 restated ). *Let $f$ be $\ell$-gradient Lipschitz. Then $r_f(\cdot, h)$ as defined in Equation 1 is $\sqrt{d}\ell$ Lipschitz for all $h \in \mathbb{R}$ and it holds that:* $\|r_f(\mathbf{x}, h) - \nabla f(\mathbf{x})\| \leq \ell\sqrt{d}|h|$

*Proof.* For the first part of the lemma we split our proof into two cases:

- For any $h \neq 0$ and any $\mathbf{x}, \mathbf{x}' \in \mathbb{R}^d$ we have

$$\|r_f(\mathbf{x}, h) - r_f(\mathbf{x}', h)\| = \left\| \sum_{l=0}^{d} \frac{f(\mathbf{x} + h\mathbf{e}_l) - f(\mathbf{x})}{h} \mathbf{e}_l - \sum_{l=0}^{d} \frac{f(\mathbf{x}' + h\mathbf{e}_l) - f(\mathbf{x}')}{h} \mathbf{e}_l \right\|$$
$$= \sqrt{\sum_{l=0}^{d} \left| \frac{f(\mathbf{x} + h\mathbf{e}_l) - f(\mathbf{x}' + h\mathbf{e}_l) - (f(\mathbf{x}) - f(\mathbf{x}'))}{h} \right|^2}$$

  Let us define the function $q_l(s) = f(\mathbf{x} + s\mathbf{e}_l) - f(\mathbf{x}' + s\mathbf{e}_l)$ for all $l \in [d]$. Then by applying the mean value theorem we get

$$\|r_f(\mathbf{x}, h) - r_f(\mathbf{x}', h)\| = \sqrt{\sum_{l=0}^{d} \left| \frac{q_l(h) - q_l(0)}{h} \right|^2} = \sqrt{\sum_{l=0}^{d} |q_l'(\xi_l)|^2}$$

  for some $\xi_l \in (0, h)$. We have that $q_l'(\xi_l) = \frac{\partial f(\mathbf{x} + \xi_l \mathbf{e}_l)}{\partial x_l} - \frac{\partial f(\mathbf{x}' + \xi_l \mathbf{e}_l)}{\partial x_l}$. If $f$ is $\ell$-gradient Lipschitz so are all the partial derivatives

$$\|r_f(\mathbf{x}, h) - r_f(\mathbf{x}', h)\| \leq \sqrt{\sum_{l=0}^{d} \ell^2 \|\mathbf{x} - \mathbf{x}'\|^2} = \sqrt{d}\ell\|\mathbf{x} - \mathbf{x}'\|$$

- For the special case of $h = 0$

$$\|r_f(\mathbf{x}, 0) - r_f(\mathbf{x}', 0)\| = \|\nabla f(\mathbf{x}) - \nabla f(\mathbf{x}')\| \leq \ell\|\mathbf{x} - \mathbf{x}'\| \leq \sqrt{d}\ell\|\mathbf{x} - \mathbf{x}'\|$$

Similarly, for the second part of the lemma we have that for any $h \neq 0$ and any $\mathbf{x}$

$$\|r_f(\mathbf{x}, h) - \nabla f(\mathbf{x})\| = \left\| \sum_{l=0}^{d} \frac{f(\mathbf{x} + h\mathbf{e}_l) - f(\mathbf{x})}{h} \mathbf{e}_l - \nabla f(\mathbf{x}) \right\|$$

$$= \sqrt{\sum_{l=0}^{d} \left| \frac{f(\mathbf{x} + h\mathbf{e}_l) - f(\mathbf{x})}{h} - \frac{\partial f(\mathbf{x})}{\partial x_l} \right|^2}$$

For each $l \in [d]$ we use the mean value theorem so that for some $x_l : |\xi_l| \leq |h|$ we have

$$\|r_f(\mathbf{x}, h) - \nabla f(\mathbf{x})\| = \sqrt{\sum_{l=0}^{d} \left| \frac{\partial f(\mathbf{x} + \xi_l \mathbf{e}_l)}{\partial x_l} - \frac{\partial f(\mathbf{x})}{\partial x_l} \right|^2}$$

$$\leq \sqrt{\sum_{l=0}^{d} (\ell \xi_l)^2} \leq \ell \sqrt{d} |h|$$

For $h = 0$ the requested inequality holds as an equality. $\qquad \square$

As noted in the main paper, recent studies have analyzed zero order optimization by carefully crafting a smoothed version of the original objective function. These arguments are also applicable to our case as well. The following lemmas show why these approaches lead $\text{poly}(d, \epsilon^{-1})$ slowdown in terms of number of iterations and function evaluations.

## A.2 Black box reductions to first order methods

Algorithm 3 of [JLGJ18], uses approximate gradient evaluations at randomly sampled points around the current iterate to get an estimate of the gradient of $f$. This estimate is then perturbed with noise in order to avoid any potential saddle point.

---
**Algorithm 3** First order Perturbed Stochastic Gradient Descent (FPSGD)
---
**Input:** $\mathbf{x}_0$, learning rate $\eta$, noise radius $r$, mini-batch size $m$.
    **for** $t = 0, 1, \ldots,$ **do**
        sample $(\mathbf{z}_t^{(1)}, \cdots, \mathbf{z}_t^{(m)}) \sim \mathcal{N}(0, \sigma^2 I)$
        $\mathbf{g}_t(\mathbf{x}_t) \leftarrow \sum_{i=1}^{m} \mathbf{g}(\mathbf{x}_t + \mathbf{z}_t^{(i)})$
        $\mathbf{x}_{t+1} \leftarrow \mathbf{x}_t - \eta(\mathbf{g}_t(\mathbf{x}_t) + \xi_t), \qquad \xi_t \text{ uniformly } \sim \mathbb{B}_0(r)$
    **end for**
    **return** $\mathbf{x}_T$
---

**Lemma 5.** *Let $f : \mathbb{R}^d \to \mathbb{R}$ be a bounded, $L$-continuous, $\ell$-gradient, $\rho$-Hessian Lipschitz function. Additionally, suppose that we have access to a function $g : \mathbb{R}^d \to \mathbb{R}$ such that $\|\nabla g - \nabla f\|_\infty \leq \nu$. Then, [JLGJ18]'s FPSG method needs $\tilde{\mathcal{O}}(\frac{d^3}{\epsilon^4})$ evaluations of $\nabla g$ to converge to an $\epsilon$-SOSP .*

*Proof.* We will show the main steps that [JLGJ18] followed in Section E of the Appendix. The first step of the proof is to define the Gaussian smoothing of function $g$ with parameter $\sigma$

$$g_\sigma(\mathbf{x}) = \mathbb{E}_{\mathbf{z} \sim \mathcal{N}(0, \sigma^2 I)} g(\mathbf{x} + \mathbf{z})$$

One can show that

$$\nabla g_\sigma(\mathbf{x}) = \mathbb{E}_{\mathbf{z} \sim \mathcal{N}(0, \sigma^2 I)} \nabla g(\mathbf{x} + \mathbf{z})$$

$$\nabla^2 g_\sigma(\mathbf{x}) = \mathbb{E}_{\mathbf{z} \sim \mathcal{N}(0, \sigma^2 I)} \nabla^2 g(\mathbf{x} + \mathbf{z})$$

Additionally Lemma 48 of [JLGJ18] tells us that the gradients and Hessians of $g_\sigma$ and $f$ are close to each other and that $g_\sigma$ is gradient Lipschitz and Hessian Lipschitz.

- $g_\sigma$ is $O(\ell + \frac{\nu}{\sigma})$ gradient Lipschitz and $O(\rho + \frac{\nu}{\sigma^2})$ Hessian Lipschitz.

- $\|\nabla g_\sigma(\mathbf{x}) - \nabla f(\mathbf{x})\| \leq \mathcal{O}(\rho d\sigma^2 + \nu)$ and $\|\nabla^2 g_\sigma(\mathbf{x}) - \nabla^2 f(\mathbf{x})\| \leq \mathcal{O}(\rho\sqrt{d}\sigma + \nu)$

Then Lemma 54 of [JLGJ18] proves that a $\frac{\epsilon}{\sqrt{d}}$-SOSP of $g_\sigma$ is also a $\mathcal{O}(\epsilon)$ stationary point of $f$ if

$$\sigma \leq \mathcal{O}(\sqrt{\frac{\epsilon}{\rho d}})$$

$$\nu \leq \mathcal{O}(\frac{\epsilon}{\sqrt{d}})$$

For the aforementioned choices of $\nu$ and $\sigma$, $\nabla g$ is bounded

$$\|\nabla g_\sigma(\mathbf{x})\| \leq \|\nabla g_\sigma(\mathbf{x}) - \nabla f(\mathbf{x})\| + \|\nabla f(\mathbf{x})\| \leq \sqrt{d}\nu + L \leq \epsilon + L$$

So $g(\mathbf{x} + \mathbf{z})$ is $\mathcal{O}(\epsilon + L)$ sub-gaussian. Notice also that by replacing with the upper bounds on $\sigma$ and $\nu$ one can observe that the Lipschitz constant of $\nabla^2 g_\sigma$ is $\mathcal{O}(\rho\sqrt{d})$. This is the main reason that a $\frac{\epsilon}{\sqrt{d}}$-SOSP of $g_\sigma$ is required.

According to Theorem 65 of [JLGJ18] getting an $\epsilon$-SOSP of $g_\sigma$ requires $\tilde{\mathcal{O}}(d/\epsilon^4)$ number of evaluations of $\nabla g$. So to get an $\frac{\epsilon}{\sqrt{d}}$-SOSP of $g_\sigma$, one would require $\tilde{\mathcal{O}}(d^3/\epsilon^4)$ number of evaluations of $\nabla g$. $\qquad\square$

Notice that the above theorem makes the technical assumption that the gradient approximator is a gradient of a function, that may not be true for standard finite differences approximators. The Lemma below for ZPSG does not have the same limitation. In contrast to FPSG, Algorithm 4 works with function evaluations directly to come up with appropriate gradient evaluations.

---

**Algorithm 4** Zero order Perturbed Stochastic Gradient Descent (ZPSGD)

---

**Input:** $\mathbf{x}_0$, learning rate $\eta$, noise radius $r$, mini-batch size $m$.
    **for** $t = 0, 1, \ldots,$ **do**
        sample $(\mathbf{z}_t^{(1)}, \cdots, \mathbf{z}_t^{(m)}) \sim \mathcal{N}(0, \sigma^2 I)$
        $\mathbf{g}_t(\mathbf{x}_t) \leftarrow \sum_{i=1}^m \mathbf{z}_t^{(i)}[f(\mathbf{x}_t + \mathbf{z}_t^{(i)}) - f(\mathbf{x}_t)]/(m\sigma^2)$
        $\mathbf{x}_{t+1} \leftarrow \mathbf{x}_t - \eta(\mathbf{g}_t(\mathbf{x}_t) + \xi_t), \qquad \xi_t$ uniformly $\sim \mathbb{B}_0(r)$
    **end for**
    **return** $\mathbf{x}_T$

---

**Lemma 6.** *Let $f : \mathbb{R}^d \to \mathbb{R}$ be a bounded, $L$-continuous, $\ell$-gradient, $\rho$-Hessian Lipschitz function. Then, [JLGJ18]'s ZPSG method needs $\tilde{\mathcal{O}}(\frac{d^2}{\epsilon^5})$ evaluations of $f$ to converge to an $\epsilon$-SOSP .*

*Proof.* We will show the main steps that [JLGJ18] followed in Section A of the Appendix. The first step of the proof is to define the Gaussian smoothing of function $f$ with parameter $\sigma$

$$f_\sigma(\mathbf{x}) = \mathbb{E}_{\mathbf{z}\sim\mathcal{N}(0,\sigma^2 I)} f(\mathbf{x} + \mathbf{z})$$

One can show that

$$\nabla f_\sigma(\mathbf{x}) = \mathbb{E}_{\mathbf{z}\sim\mathcal{N}(0,\sigma^2 I)} \nabla f(\mathbf{x} + \mathbf{z})$$

$$\nabla^2 f_\sigma(\mathbf{x}) = \mathbb{E}_{\mathbf{z}\sim\mathcal{N}(0,\sigma^2 I)} \nabla^2 f(\mathbf{x} + \mathbf{z})$$

Additionally Lemma 18 of [JLGJ18] for $\nu = 0$, tells us that the gradients and Hessians of $f_\sigma$ and $f$ are close to each other and that $f_\sigma$ is gradient Lipschitz and Hessian Lipschitz.

- $f_\sigma$ is $O(\ell)$ gradient Lipschitz and $O(\rho)$ Hessian Lipschitz.

- $\|\nabla f_\sigma(\mathbf{x}) - \nabla f(\mathbf{x})\| \leq \mathcal{O}(\rho d\sigma^2)$ and $\|\nabla^2 f_\sigma(\mathbf{x}) - \nabla^2 f(\mathbf{x})\| \leq \mathcal{O}(\rho\sqrt{d}\sigma)$

Based on this we can see that an $\epsilon$-SOSP of $f_\sigma$ is also a $\mathcal{O}(\epsilon)$ stationary point of $f$ if

$$\sigma \leq \mathcal{O}\left(\sqrt{\frac{\epsilon}{\rho d}}\right)$$

We also need to develop a random gradient approximator of $\nabla f_\sigma$ given only evaluations $f$. Based on Lemma 19

$$\nabla f_\sigma(\mathbf{x}) = \mathbb{E}_{\mathbf{z} \sim \mathcal{N}(0, \sigma^2 I)} \mathbf{z} \frac{f(\mathbf{x} + \mathbf{z}) - f(\mathbf{x})}{\sigma^2}$$

Let us define

$$g(\mathbf{x}; \mathbf{z}) = \mathbf{z} \frac{f(\mathbf{x} + \mathbf{z}) - f(\mathbf{x})}{\sigma^2}$$

Lemma 24 shows that $g$ is $\frac{B}{\sigma}$ subgaussian where $B$ is the upper bound on $|f(\mathbf{x})|$ (it exists since $f$ is bounded). Replacing with the upper bound on $\sigma$, it turns out that $g$ is $\mathcal{O}(B\sqrt{\frac{\epsilon}{\rho d}})$ subgaussian. This dependence on $d$ and $\epsilon$ is the main reason of the slowdown in this case.

According to Theorem 65 getting an $\epsilon$-SOSP of $f_\sigma$ requires $\tilde{\mathcal{O}}(d^2/\epsilon^5)$ number of evaluations of $g$. Each evaluation of $g$ requires 2 evaluations of $f$. □

In the next section, we show the complete proof of our first main result. We will use the Stable Manifold Theorem (SMT) to prove that zero-order approximate gradient descent (AGD) avoids strict saddle points.

# B   Approximate Gradient Descent Detailed proofs

Our first two lemmas prove the equivalence between the first order stationary points of $f$ and the fixed points of the AGD. Additionally we show that saddle points of the objective function correspond exactly to the unstable fixed of the proposed zero order method. Finally we show that for sufficiently small size-step the dynamical system is diffeomorphism. This critical property will allow us to generalize the consequences of SMT from a local region around a saddle point to the global domain.

## B.1   Avoiding strict saddle points

**Lemma 7.** *Assume that $g_0$ is an $(L, B, c)$ well behaved function. If $\beta < \frac{1}{B}$ and $\eta < \frac{1}{L}$ for every strict saddle point $\mathbf{x}^*$ of $f$ and we have that $\begin{pmatrix} \mathbf{x}^* \\ 0 \end{pmatrix}$ is not a stable fixed point of $g_0$. Additionally, these are the only unstable fixed points of $g_0$.*

*Proof.* For $h = 0$ and at a strict saddle $\mathbf{x}^*$, we will calculate the general differential of $g_0$.

$$\mathrm{D}g_0 \begin{pmatrix} \mathbf{x}^* \\ 0 \end{pmatrix} = \begin{pmatrix} I - \eta D_x q_x(\mathbf{x}^*, 0) & -\eta D_h q_x(\mathbf{x}^*, 0) \\ 0 & \beta \frac{\partial q_h(0)}{\partial h} \end{pmatrix}$$
$$= \begin{pmatrix} I - \eta \nabla^2 f(\mathbf{x}^*) & -\eta D_h q_x(\mathbf{x}^*, 0) \\ 0 & \beta \frac{\partial q_h(0)}{\partial h} \end{pmatrix}$$

with eigenvalues $\beta \frac{\partial q_h(0)}{\partial h}, (1 - \eta \lambda_i)$ , where $\lambda_i$ are eigenvalues of $\nabla^2 f(\mathbf{x}^*)$. Since $\mathbf{x}^*$ is a strict saddle, then there is at least one eigenvalue $\lambda_i < 0$, and $1 - \eta \lambda_i > 1$. Thus $\begin{pmatrix} \mathbf{x}^* \\ 0 \end{pmatrix}$ is an unstable fixed point of $g_0$. To prove that these are the only unstable fixed points, observe that $\beta \frac{\partial q_h(0)}{\partial h} \in (0, 1)$ so the only way $\mathrm{D}g_0 \begin{pmatrix} \mathbf{x}^* \\ 0 \end{pmatrix}$ has an eigenvalue greater than 1 is for some $\lambda_i$ to be negative and therefore $\mathbf{x}^*$ should be a strict saddle. $\qquad \square$

For the sake of completeness here we provide an extra lemma that proves the equivalence between the first order stationary points of $f$ and the fixed points of $g_0$.

**Lemma 8.** *Assume that $g_0$ is an $(L, B, c)$-well-behaved function for a function $f$ with $\beta < \frac{1}{B}$. Then for each first order stationary point of $f$ $\mathbf{x}^*$, $\begin{pmatrix} \mathbf{x}^* \\ 0 \end{pmatrix}$ is a fixed point of $g_0$. Additionally $g_0$ has no other fixed points.*

*Proof.* For $\beta < \frac{1}{B}$ we have that $g_h = \beta q_h(h)$ is a contraction since its Lipschitz constant is less than one. So the only fixed point of $g_h$ is 0. Therefore for $h \neq 0$ no point $\begin{pmatrix} \mathbf{x} \\ h \end{pmatrix}$ is a stable point. Now for $h = 0$ we get that $q_x(\mathbf{x}, h) = \nabla f(\mathbf{x})$ so we have

$$\mathbf{x}_{k+1} = \mathbf{x}_k - \eta \nabla f(\mathbf{x}_k) \tag{1}$$

So $\mathbf{x}$ is a fixed point if and only if $\nabla f(\mathbf{x}) = 0$. Combining this with the requirement that all fixed points of $g_0$ have $h = 0$ proves the lemma. $\qquad \square$

In order to prove Theorem 1 we also have to prove the diffeomorphism property of $g_0$.

**Lemma 9.** *If $g_0$ is an $(L, B, c)$ well behaved function and $\eta < \frac{1}{L}$, then $\det(\mathrm{D}g_0(\cdot)) \neq 0$.*

*Proof.* Let

$$\mathcal{K} = \mathrm{D}_x q_x(\mathbf{x}, h) \tag{2}$$

By straightforward calculation

$$\mathrm{D}g_0\begin{pmatrix} x \\ h \end{pmatrix} = \begin{pmatrix} I - \eta\mathcal{K} & -\eta\mathrm{D}_h q_x(\mathbf{x}, h) \\ 0 & \beta\frac{\partial q_h(h)}{\partial h} \end{pmatrix}$$

Given that $g(\cdot, h)$ is $L$-Lipschitz for all $h \in \mathbb{R}$, we have that $\|\mathcal{K}\|_2 \leq L$. Clearly we have that $\det(I - \eta\mathcal{K}) \neq 0$ since $\|I - \eta\mathcal{K}\|_2 \geq 1 - \eta L > 0$. Finally we have that

$$\det(\mathrm{D}g_0\begin{pmatrix} x \\ h \end{pmatrix}) = \beta\frac{\partial q_h(h)}{\partial h}\det(I - \eta\mathcal{K}) \neq 0.$$

$\square$

A straightforward application of result of [LPP$^+$19] and SMT will yields a saddle-avoidance lemma following kind :

*Let $X_f^*$ be the set of the strict saddle points of $f$, $\eta < \frac{1}{L}$ and $\beta < \frac{1}{B}$. Then it holds:*
$$\Pr\left(\left\{\begin{pmatrix}\mathbf{x}_0 \\ h_0\end{pmatrix} : \lim_{k\to\infty} \mathbf{x}_k \in X_f^*\right\}\right) = 0.$$

Notice that the random choice would be both on $\mathbf{x}_0, h_0$. **In the following subsection we will prove that a stronger result where the random initialization refers only to the $\mathbf{x}_0$'s domain is surprisingly possible via a new refinement of SMT**:

$$\forall h_0 \in \mathbb{R} : \quad \Pr(\lim_{k\to\infty} \mathbf{x}_k = \mathbf{x}^*) = 1$$

Let us first describe our general strategy for proving this refinement:

1. We will restate the Stable Manifold Theorem and understand its implications. (Section B.2.1)

2. We will study the structure of the eigenvalues of $\mathrm{D}g_0$ at fixed points of $g_0$. (Section B.2.2)

3. We will show how this affects the projections to the stable and unstable eignespaces of $\mathrm{D}g_0$. (Section B.2.3)

4. Finally we will see how this enables us to study the dimension of the stable manifold when $h_0$ is fixed. (Section B.2.4)

## B.2 A Refinement of the Stable Manifold Theorem

### B.2.1 Understanding the Stable Manifold Theorem

**Theorem 5** (Theorem III.2 & III.7 of [Shu87]). *Let $\mathbf{p}$ be a fixed point for the $\mathcal{C}^r$ local diffeomorphism $h : U \to R^n$ where $U \subset R^n$ is an open neighborhood of $p$ in $R^n$ and $r \geq 1$. Let $E_s \oplus E_c \oplus E_u$ be the invariant splitting of $R^n$ into generalized eigenspaces of $Dh(p)$ corresponding to eigenvalues of absolute value less than one, equal to one, and greater than one. To the $Dh(p)$ invariant subspace $E_s \oplus E_c$ there is an associated local $h$ invariant embedded disc $W_{sc}^{loc}$ which is the graph of a $\mathcal{C}^r$ function $r : E_s \oplus E_c \to E_u$, and ball $B$ around $p$ such that:*

$$h(W_{sc}^{loc}) \cap B \subset W_{sc}^{loc}. \text{ If } h^n(\mathbf{x}) \in B \text{ for all } n \geq 0, \text{ then } \mathbf{x} \in W_{sc}^{loc}$$

We will give some intuition on how the Stable Manifold Theorem restricts the dimensionality of the stable manifold. It essentially boils down to restricting the dimensionality of the manifold $W_{sc}^{loc}$. Let us have a $\mathbf{x} \in U$, then this can be decomposed in two vectors $\mathbf{x}_{sc}$ and $\mathbf{x}_u$, the projection of $x$ to $E_s \oplus E_c$ and $E_u$ respectively. Thus by the construction of $W_{sc}^{loc}$ in the proof of the Stable Manifold theorem, we know that there is a function $r : E_s \oplus E_c \to E_u$ such that if $\mathbf{x} \in W_{sc}^{loc}$ then $(\mathbf{x}_{sc}, \mathbf{x}_u) \in graph(r)$, or equivalently it holds that $\mathbf{x}_u = r(\mathbf{x}_{sc})$. By the construction of $r$, $r$ is smooth so now $dim(W_{sc}^{loc}) = dim(\mathrm{graph}(r)) = dim(E_s \oplus E_c)$. To understand why the last statement is true, the interested reader can look at example 5.14 of [Lor08].

### B.2.2 Eigenvalues of the Jacobian at fixed points

Our main tool for understanding the structure of the eigenvalues of $\mathrm{D}g_0$ at fixed points of $g_0$ is comparing it and contrasting it with its first order counterpart, gradient descent. Here is the dynamical system of gradient descent:

$$\mathbf{x}_{k+1} = g_1(\mathbf{x}_k) = \mathbf{x}_k - \eta \nabla f(\mathbf{x}_k)$$

Now let us pick a fixed point of $f$, $\mathbf{x}^*$. Then

$$\mathrm{D}g_1(\mathbf{x}^*) = I - \eta \nabla^2 f(\mathbf{x}^*)$$

is a symmetrical matrix for the $C^2$ function $f$. Then we can write down its real orthonormal eigenvectors $\{\mathbf{v}_i\}_{i=1}^d$. Without loss of generality we can reorder them so that the $k$ first eigenvectors correspond to eigenvalues less than one, the next $s$ correspond to eigenvalues that are equal to one and and the last ones correspond to eigenvalues that are larger than one in absolute value. Based on this separation between the eigenvectors, we can now define the following three vector spaces

$$\begin{aligned} E_s^{g_1} &= [\{\mathbf{v}_1, \cdots, \mathbf{v}_k\}] \\ E_c^{g_1} &= [\{\mathbf{v}_{k+1}, \cdots, \mathbf{v}_{k+s}\}] \\ E_u^{g_1} &= [\{\mathbf{v}_{k+s+1}, \cdots, \mathbf{v}_d\}] \end{aligned}$$

Then we can prove the following interesting lemma

**Lemma 10.** *If $\mathbf{v}$ is eigenvector of $Dg_1(x^*)$ then $\binom{\mathbf{v}}{0}$ is eigenvector of $Dg_0\binom{\mathbf{x}^*}{0}$ with the same eigenvalue.*

*Proof.* By straightforward calculation

$$\begin{aligned} \mathrm{D}g_0\begin{pmatrix}\mathbf{x}^* \\ 0\end{pmatrix} &= \begin{pmatrix} I - \eta D_x q_x(\mathbf{x}^*, 0) & -\eta D_h q_x(\mathbf{x}^*, 0) \\ 0 & \beta \frac{\partial q_h(0)}{\partial h} \end{pmatrix} \\ &= \begin{pmatrix} I - \eta \nabla^2 f(\mathbf{x}^*) & -\eta D_h q_x(\mathbf{x}^*, 0) \\ 0 & \beta \frac{\partial q_h(0)}{\partial h} \end{pmatrix} \\ &= \begin{pmatrix} \mathrm{D}g_1(\mathbf{x}^*) & -\eta D_h q_x(\mathbf{x}^*, 0) \\ 0 & \beta \frac{\partial q_h(0)}{\partial h} \end{pmatrix} \end{aligned}$$

Indeed if $\mathbf{v}$ is eigenvector of $Dg_1(\mathbf{x}^*)$ with eigenvalue $\lambda$ then

$$\mathrm{D}g_0\begin{pmatrix}\mathbf{x}^* \\ 0\end{pmatrix}\begin{pmatrix}\mathbf{v} \\ 0\end{pmatrix} = \begin{pmatrix} \mathrm{D}g_1(\mathbf{x}^*) & -\eta D_h q_x(\mathbf{x}^*, 0) \\ 0 & \beta \frac{\partial q_h(0)}{\partial h} \end{pmatrix}\begin{pmatrix}\mathbf{v} \\ 0\end{pmatrix} = \begin{pmatrix}\lambda\mathbf{v} \\ 0\end{pmatrix} = \lambda\begin{pmatrix}\mathbf{v} \\ 0\end{pmatrix}$$

□

Now we now the form of the $d$ out of the $d+1$ generalized eigenvalues of $\mathrm{D}g\binom{\mathbf{x}^*}{0}$. There must be at least one more generalized eigenvector along with its corresponding eigenvalue. It is known that generalized eigenvectors span the whole space. But so far all the eigenvectors have a zero in the last coordinate. So the last generalized eigenvector must have a non-zero value in the last coordinate. Without loss of generality we can assume that the last coordinate is 1. So the vector will be of the form $\binom{\check{\mathbf{v}}}{1}$. We would like to determine its corresponding eigenvalue.

**Lemma 11.** *The eigenvalue of $\mathrm{D}g_0\binom{\mathbf{x}^*}{0}$ that corresponds to $\binom{\check{\mathbf{v}}}{1}$ is $\beta\frac{\partial q_h(0)}{\partial h}$*

*Proof.* Since the last row of $\mathrm{D}g_0\binom{\mathbf{x}^*}{0}$ contains only one non-zero element, we know that the characteristic polynomial $p_0$ of $\mathrm{D}g$ can be written as

$$\det(\mathrm{D}g_0\begin{pmatrix}\mathbf{x}^* \\ 0\end{pmatrix} - \lambda I_{d+1\times d+1}) = \det(\mathrm{D}g_1(\mathbf{x}^*) - \lambda I_{d\times d})\det(\beta\frac{\partial q_h(0)}{\partial h} - \lambda)$$

Given that all the other eigenvalues cover the roots of the first term, we know that the last eigenvalue is $\beta\frac{\partial q_h(0)}{\partial h}$.

□

By assumption we know that $0 < \beta\frac{\partial q_h(0)}{\partial h} < 1$. Thus the last generalized eigenvector corresponds to a stable eigenvalue. Now we can write down the following

$$E_s^g = \left[\left\{\begin{pmatrix}\mathbf{v}_1\\0\end{pmatrix}, \cdots, \begin{pmatrix}\mathbf{v}_k\\0\end{pmatrix}, \begin{pmatrix}\tilde{\mathbf{v}}\\1\end{pmatrix}\right\}\right]$$

$$E_c^g = \left[\left\{\begin{pmatrix}\mathbf{v}_{k+1}\\0\end{pmatrix}, \cdots, \begin{pmatrix}\mathbf{v}_{k+s}\\0\end{pmatrix}\right\}\right] \tag{3}$$

$$E_u^g = \left[\left\{\begin{pmatrix}\mathbf{v}_{k+s+1}\\0\end{pmatrix}, \cdots, \begin{pmatrix}\mathbf{v}_d\\0\end{pmatrix}\right\}\right]$$

### B.2.3  Projections to stable and unstable eigenspaces of the Jacobian

In this paragraph we want to learn more about the projection to the stable and unstable eigenspaces of $Dg$. Specifically for any vector $\begin{pmatrix}\mathbf{x}\\h\end{pmatrix}$, there are unique $\mathbf{x}_{sc}^{g_0}$, $\mathbf{x}_u^{g_0}$, $h_s$, $h_u$ such that

$$\begin{pmatrix}\mathbf{x}\\h\end{pmatrix} = \begin{pmatrix}\mathbf{x}_{sc}^{g_0}\\h_s\end{pmatrix} + \begin{pmatrix}\mathbf{x}_u^{g_0}\\h_u\end{pmatrix}$$

$$\begin{pmatrix}\mathbf{x}_{sc}^{g_0}\\h_s\end{pmatrix} \in E_s^{g_0} \oplus E_c^{g_0} \text{ and } \begin{pmatrix}\mathbf{x}_u^{g_0}\\h_u\end{pmatrix} \in E_u^{g_0}$$

Let us compute these projections. Given that the generalized eigenvectors span the whole space, we have that there are unique $\lambda_i \in \mathbb{R}$ such that

$$\begin{pmatrix}\mathbf{x}\\h\end{pmatrix} = \sum_{i=1}^n \lambda_i \begin{pmatrix}\mathbf{v}_i\\0\end{pmatrix} + \lambda_{n+1}\begin{pmatrix}\tilde{\mathbf{v}}\\1\end{pmatrix} \Leftrightarrow$$

$$\lambda_{n+1} = h \text{ and } \mathbf{x} = \sum_{i=1}^n \lambda_i \mathbf{v}_i + h\tilde{\mathbf{v}} \Leftrightarrow$$

$$\lambda_{n+1} = h \text{ and } \mathbf{x} - h\tilde{\mathbf{v}} = \sum_{i=1}^n \lambda_i \mathbf{v}_i \Leftrightarrow$$

$$\lambda_{n+1} = h \text{ and } \lambda_i = \langle \mathbf{x} - h\tilde{\mathbf{v}}, \mathbf{v}_i\rangle$$

Since $\mathbf{v}_i$ are orthogonal as eigenvectors of a symmetrical matrix. We can now find the vectors and values $\mathbf{x}_{sc}^{g_0}$, $\mathbf{x}_u^{g_0}$, $h_s$, $h_u$

$$\mathbf{x}_{sc}^{g_0} = \sum_{i=1}^{k+\ell} \lambda_i \mathbf{v}_i + h\tilde{\mathbf{v}}$$

$$= \sum_{i=1}^{k+\ell} \langle \mathbf{x} - h\tilde{\mathbf{v}}, \mathbf{v}_i\rangle \mathbf{v}_i + h\tilde{\mathbf{v}}$$

$$= \sum_{i=1}^{k+\ell} \langle \mathbf{x}, \mathbf{v}_i\rangle \mathbf{v}_i + h\left(\tilde{\mathbf{v}} - \sum_{i=1}^{k+\ell} \langle \tilde{v}, \mathbf{v}_i\rangle \mathbf{v}_i\right)$$

$$\mathbf{x}_u^{g_0} = \sum_{i=k+\ell+1}^n \lambda_i \mathbf{v}_i$$

$$= \sum_{i=k+\ell+1}^n \langle \mathbf{x} - h\tilde{\mathbf{v}}, \mathbf{v}_i\rangle \mathbf{v}_i$$

$$= \sum_{i=k+\ell+1}^n \langle \mathbf{x}, \mathbf{v}_i\rangle \mathbf{v}_i - h\sum_{i=k+\ell+1}^n \langle \tilde{\mathbf{v}}, \mathbf{v}_i\rangle \mathbf{v}_i$$

$$h_s = h \text{ and } h_u = 0$$

Once again we will compare and contrast with the first order case. Equivalently for every vector $\mathbf{x}$ there are unique $\mathbf{x}_{sc}^{g_1}$, $\mathbf{x}_u^{g_1}$ such that

$$\mathbf{x} = \mathbf{x}_{sc}^{g_1} + \mathbf{x}_u^{g_1}$$

$$\mathbf{x}_{sc}^{g_1} \in E_s^{g_1} \oplus E_c^{g_1} \text{ and } \mathbf{x}_u^{g_1} \in E_u^{g_1}$$

Let us define

$$\mathbf{q} = \tilde{\mathbf{v}} - \sum_{i=1}^{k+\ell} \langle \tilde{\mathbf{v}}, \mathbf{v}_i \rangle \mathbf{v}_i$$

$$= \sum_{i=k+\ell+1}^{n} \langle \tilde{\mathbf{v}}, \mathbf{v}_i \rangle \mathbf{v}_i \tag{4}$$

Then clearly

$$\begin{aligned}
\mathbf{x}_{sc}^{g_0} &= \mathbf{x}_{sc}^{g_1} + h\mathbf{q} \\
\mathbf{x}_{u}^{g_0} &= \mathbf{x}_{u}^{g_1} - h\mathbf{q} \\
h_{sc} &= h \\
h_u &= 0
\end{aligned} \tag{5}$$

### B.2.4 Restricting the dimension of the stable manifold for fixed initial $h$

In this paragraph we are ready to finally prove Theorem 1.

**Theorem 6** (Theorem 1 restated)**.** *Let $g_0$ be a $(L, B, c)$-well-behaved function for function $f$. Let $X_f^*$ be the set of strict saddle points of $f$. Then if $\eta < \frac{1}{L}$ and $\beta < \frac{1}{B}$:*

$$\forall h_0 \in \mathbb{R} : \mu(\{\mathbf{x}_0 : \lim_{k \to \infty} \mathbf{x}_k \in X_f^*\}) = 0$$

*Proof.* Without loss of generality let us have a fixed $h = h_0$. Let us define $M_{h_0}$ as

$$M_{h_0} = \{\mathbf{x}_0 \in \mathbb{R}^n : \lim_{k \to \infty} g_0^k(\mathbf{x}_0, h_0) = (\mathbf{x}^*, 0) \text{ and } \mathbf{x}^* \in X_f^*\}$$

We want to prove that the set $M$ has measure 0. Let us apply the Stable Manifold Theorem on $g_0$ for all fixed points $\mathbf{p} = (\mathbf{x}^*, 0) \in X_f^* \times \{0\}$. Let $B_\mathbf{p}, W_{sc,\mathbf{p}}^{loc}$ be the ball and the corresponding manifold derived by Theorem 5. We consider the union of those balls $\mathcal{B} = \bigcup B_p$. The following property for $\mathbb{R}^N$ holds:

**Theorem** (Lindelöfs lemma)**.** *For every open cover there is a countable subcover.*

Therefore due to Lindelöfs lemma, we can find a countable subcover for $\mathcal{B}$, i.e., there exists a countable family of fixed-points $\mathbf{p}_0, \mathbf{p}_1, \cdots$ such that $\mathcal{B} = \bigcup_{m=0}^{+\infty} B_{p_m}$. Once again, based on Theorem 5, if starting from $\mathbf{x}_0$ one converges to an unstable fixed point then it holds that

$$\begin{aligned}
\mathbf{x}_0 \in M_{h_0} &\Rightarrow \exists m, t_0 : \forall t \geq t_0 \ (\mathbf{x}_t, h_t) = g_0^t(\mathbf{x}_0, h_0) \text{ and } (\mathbf{x}_t, h_t) \in B_{\mathbf{p}_m} \\
&\Rightarrow \exists m, t_0 : (\mathbf{x}_{t_0}, h_{t_0}) = g_0^{t_0}(\mathbf{x}_0, h_0) \text{ and } (\mathbf{x}_{t_0}, h_{t_0}) \in W_{sc,\mathbf{p}_m}^{loc}
\end{aligned}$$

Let us define

$$U_t^m = \{\mathbf{x}_0 \in \mathbb{R}^d : (\mathbf{x}_t, h_t) = g_0^t(\mathbf{x}_0, h_0) \text{ and } (\mathbf{x}_t, h_t) \in W_{sc,\mathbf{p}_m}^{loc}\}$$

Therefore we have

$$M_{h_0} \subseteq \bigcup_{m=0}^{\infty} \bigcup_{t=0}^{\infty} U_t^m$$

Now it suffices to prove that all $U_t^m$ sets have zero measure. Let us first prove the following lemma as a stepping stone.

**Lemma.** *Let us define the following set of points*

$$R_h^m = \{\mathbf{x} \in \mathbb{R}^d : (\mathbf{x}, h) \in W_{sc,\mathbf{p}_m}^{loc}\}$$

*Then $dim(R_h^m) < d$.*

*Proof.* Based on our discussion on the Stable Manifold Theorem, we know that there is a smooth function $r : E_s^g \oplus E_c^g \to E_u^g$ such that

$$\begin{pmatrix} \mathbf{x} \\ h \end{pmatrix} \in W_{sc,\mathbf{p}_m}^{loc} \Rightarrow \begin{pmatrix} \mathbf{x}_u^{g_0} \\ h_s \end{pmatrix} = r(\mathbf{x}_{sc}^{g_0}, h_u)$$

where $\mathbf{x}_u^g$, $\mathbf{x}_{sc}^g$, $h_s$ and $h_u$ the components of the projections to $E_s^{g_0} \oplus E_c^{g_0}$ and $E_u^{g_0}$ as defined in the Equations of 3. Now using our analysis in the Equations of 5

$$\begin{pmatrix} \mathbf{x} \\ h \end{pmatrix} \in W_{sc,\mathbf{p}_m}^{loc} \Rightarrow \begin{pmatrix} \mathbf{x}_u^{g_1} - h\mathbf{q} \\ 0 \end{pmatrix} = r(\mathbf{x}_{sc}^{g_1} + h\mathbf{q}, h)$$

where $\mathbf{q}$ is the vector we defined in Equation 4. Let $\prod$ be the projection that for each $\begin{pmatrix} \mathbf{x} \\ h \end{pmatrix} \in \mathbb{R}^{d+1}$ returns $\mathbf{x}$. Then we can define the following smooth function

$$r_h' : E_s^{g_1} \oplus E_c^{g_1} \to E_u^{g_1} \ r_h'(\mathbf{x}) = h\mathbf{q} + \prod r(\mathbf{x} + h\mathbf{q}, h).$$

Using the $\{\mathbf{v}_i\}_{i=1}^n$ as a basis we can write

$$\begin{pmatrix} \mathbf{x} \\ h \end{pmatrix} \in W_{sc,\mathbf{p}_m}^{loc} \Rightarrow \mathbf{x}_u^{g_1} = r_h'(\mathbf{x}_{sc}^{g_1}) \Rightarrow \mathbf{x} \in \text{graph}(r_h')$$

Therefore $dim(R_h^m) \leq dim(E_s^{g_1} \oplus E_c^{g_1}) < d$ since $\mathbf{p}_m$ corresponds to an unstable fixed point of $g_1$. $\qquad \square$

Then we can prove the following lemma

**Lemma 12.** *The measure of $U_t^m$ is zero.*

*Proof.* We will do this by contradiction. Let us assume that $U_t$ has non-zero measure. Let us define

$$W_0^m = \{\mathbf{x} \in \mathbb{R}^n : x \in U_t^m\}$$
$$W_1^m = \{\mathbf{x} \in \mathbb{R}^n : x \in g_0(W_1^m, h_1)\}$$
$$\vdots$$
$$W_t^m = \{\mathbf{x} \in \mathbb{R}^n : x \in g_0(W_{t-1}^m, h_{t-1})\}$$

Given that $g(\cdot, h_i)$ is a diffeomorphism for all $i$, we have that $W_i$ has non zero measure. Observe that

$$W_t^m \subseteq R_{h_t}^m$$

and so $dim(W_t^m) < d$ and $W_t^m$ has measure zero leading to a contradiction. $\qquad \square$

Since the countable union of zero measure sets is zero measure we clearly have that $M_{h_0}$ has measure zero as requested. $\qquad \square$

In the previous section, we provided sufficient conditions to avoid convergence to strict saddle points. These results are meaningful however only if $\lim_{k \to \infty} \mathbf{x}_k = \mathbf{x}^*$. Thus in order to complete the proof of 3, in the following section we will provide sufficient conditions such that the dynamic system of AGD converges.

## B.3 Convergence

We will refer to the error of the gradient approximation as

$$\varepsilon_k = q_x(\mathbf{x}_k, h_k) - \nabla f(\mathbf{x}_k).$$

In order to prove the convergence firstly we establish a lower bound for the decrease of the function that is connected with the norm of the gradient and its approximation error (Lemma 2). We also prove that our scheme yields to an exponential decrease of that error (Lemma 14). Given those lemmas we can prove an exact and an $\epsilon-$first order stationary convergence theorem.

**Lemma 13** (Lemma 2 restated). *Suppose that $g_0$ is a $(L, B, c)$-well-behaved function for a $\ell$-gradient Lipschitz function $f$. If $\eta \leq \frac{1}{\ell}$ then we have that*

$$f(\mathbf{x}_{k+1}) \leq f(\mathbf{x}_k) - \frac{\eta}{2}\left(\|\nabla f(\mathbf{x}_k)\|^2 - \|\varepsilon_k\|^2\right) \tag{6}$$

*Proof.*

$$f(\mathbf{x}_{k+1}) \leq f(\mathbf{x}_k) + \nabla f(\mathbf{x}_k)^\top (\mathbf{x}_{k+1} - \mathbf{x}_k) + \frac{\ell}{2}\|\mathbf{x}_{k+1} - \mathbf{x}_k\|^2$$

$$\leq f(\mathbf{x}_k) - \eta \nabla f(\mathbf{x}_k)^\top q_x(\mathbf{x}_k, h_k) + \frac{\eta^2\ell}{2}\|q_x(\mathbf{x}_k, h_k)\|^2$$

$$\leq f(\mathbf{x}_k) - \eta \nabla f(\mathbf{x}_k)^\top (\nabla f(\mathbf{x}_k) + \varepsilon_k) + \frac{\eta^2\ell}{2}\|\nabla f(\mathbf{x}_k) + \varepsilon_k\|^2$$

$$\leq f(\mathbf{x}_k) - \eta \nabla f(\mathbf{x}_k)^\top (\nabla f(\mathbf{x}_k) + \varepsilon_k) + \frac{\eta}{2}\|\nabla f(\mathbf{x}_k) + \varepsilon_k\|^2$$

$$\leq f(\mathbf{x}_k) - \frac{\eta}{2}\left(\|\nabla f(\mathbf{x}_k)\|^2 - \|\varepsilon_k\|^2\right)$$

$\square$

**Lemma 14** (Exponentially Decreasing $\varepsilon_k$). *Suppose that $g_0$ is a $(L, B, c)$-well-behaved function for a function $f$. Then we have that*

$$\|\varepsilon_k\| \leq c|h_0|(\beta B)^k$$

*Proof.* Since $q_h$ is $B$-Lipschitz

$$|h_{k+1}| = |\beta q_h(h_k) - \beta q_h(0)| \leq \beta B|h_k|$$

Therefore we have that

$$|h_k| \leq (\beta B)^k|h_0|$$

Based on property 3 of the $(L, B, c)$-well-behaved function we have that

$$\|\varepsilon_k\| = \|q_x(\mathbf{x}_k, h_k) - \nabla f(\mathbf{x}_k)\| \leq c|h_k| = (\beta B)^k|h_0|$$

$\square$

Now we are ready to start our proof for the convergence to the first order stationary points.

**Theorem 7** ( Theorem 2 Restated). *Suppose that $g_0$ is a $(L, B, c)$-well-behaved gradient function for a $\ell$-gradient Lipschitz function $f$. Let $\eta \leq \frac{1}{\ell}$, $\beta < \frac{1}{B}$. Then if $f$ is lower bounded*

$$\lim_{k \to \infty} \|\nabla f(\mathbf{x}_k)\| = 0$$

*Proof.* Applying Lemma 2 repeatedly we get

$$f(\mathbf{x}_0) - f(\mathbf{x}_k) \geq \frac{\eta}{2} \sum_{i=0}^{k} \left( \|\nabla f(\mathbf{x}_i)\|^2 - \|\varepsilon_i\|^2 \right)$$

We now have that

$$f(\mathbf{x}_0) - f(\mathbf{x}_k) + \frac{\eta}{2} \sum_{i=0}^{k} \|\varepsilon_i\|^2 \geq \frac{\eta}{2} \sum_{i=0}^{k} \|\nabla f(\mathbf{x}_i)\|^2$$

$$f(\mathbf{x}_0) - f(\mathbf{x}_k) + \frac{\eta}{2} \sum_{i=0}^{\infty} \|\varepsilon_i\|^2 \geq \frac{\eta}{2} \sum_{i=0}^{k} \|\nabla f(\mathbf{x}_i)\|^2$$

$$f(\mathbf{x}_0) - f(\mathbf{x}_k) + \frac{\eta}{2} \sum_{i=0}^{\infty} \left( c|h_0|(\beta B)^i \right)^2 \geq \frac{\eta}{2} \sum_{i=0}^{k} \|\nabla f(\mathbf{x}_i)\|^2$$

$$f(\mathbf{x}_0) - f(\mathbf{x}_k) + \frac{\eta}{2} \frac{c^2 h_0^2}{1 - (\beta B)^2} \geq \frac{\eta}{2} \sum_{i=0}^{k} \|\nabla f(\mathbf{x}_i)\|^2$$

Given that $f$ is lower bounded, $f(\mathbf{x}_0) - f(\mathbf{x}_k)$ and therefore the whole left hand side is upper bounded which means the series sum in the right hand side is upper bounded. Since this is a series of non negative terms this means that the series converges and therefore

$$\lim_{k \to \infty} \|\nabla f(\mathbf{x}_k)\| = 0$$

□

For the sake of completeness, we will analyze the convergence rate to $\epsilon$-first order stationary points in this setting. This would enable us to to make a fair comparison with previous results that assume a fixed $h_k = h_0$. Notice that the following result improves over previous work in randomized zero order gradient approximations. In [NS17], it was proved that using a randomized oracle that requires 2 function evaluations per iteration, one could get an in expectation $\epsilon$-first order stationary point after $\mathcal{O} \left( d\ell \left( f(\mathbf{x}_0) - f^* \right) / \epsilon^2 \right)$ iterations. For the case of $q_x$ using $r_f$ as defined in Equation 1 of the Section 3, we have just proved that with $d + 1$ function evaluations per iteration we can get a $\epsilon$-first order stationary point after only $\mathcal{O} \left( \ell \left( f(\mathbf{x}_0) - f^* \right) / \epsilon^2 \right)$ iterations. Thus for the same number of function evaluations up to constants, our work provides deterministic guarantees whereas [NS17] provides guarantees only in expectation.

**Theorem 8** ($\epsilon$-first order stationary points). *Suppose that $g_0$ is a $(L, B, c)$-well-behaved gradient function for a $\ell$-gradient Lipschitz function $f$. Let $q_h(h) = h$ and $\beta = 1$, $\eta = \frac{1}{\ell}$. Then if $f$ has minimum value $f^*$ and $h_0 = \frac{\epsilon}{\sqrt{2}c}$, the required number of iterations to reach a $\epsilon$-first order stationary point is*

$$\mathcal{O} \left( \frac{\ell \left( f(\mathbf{x}_0) - f^* \right)}{\epsilon^2} \right)$$

*Proof.* Applying Lemma 2 repeatedly we get

$$f(\mathbf{x}_0) - f(\mathbf{x}_k) \geq \frac{1}{2\ell} \sum_{i=0}^{k} \left( \|\nabla f(\mathbf{x}_i)\|^2 - \|\varepsilon_i\|^2 \right)$$

We now have that

$$f(\mathbf{x}_0) - f(\mathbf{x}_k) + \frac{1}{2\ell}\sum_{i=0}^{k}\|\varepsilon_i\|^2 \geq \frac{1}{2\ell}\sum_{i=0}^{k}\|\nabla f(\mathbf{x}_i)\|^2$$

$$f(\mathbf{x}_0) - f(\mathbf{x}_k) + \frac{k+1}{2\ell}\left(c|h_0|\right)^2 \geq \frac{1}{2\ell}\sum_{i=0}^{k}\|\nabla f(\mathbf{x}_i)\|^2$$

$$\frac{\ell(f(\mathbf{x}_0) - f(\mathbf{x}_k))}{2(k+1)} + c^2|h_0|^2 \geq \frac{1}{k+1}\sum_{i=0}^{k}\|\nabla f(\mathbf{x}_i)\|^2$$

$$\frac{\ell(f(\mathbf{x}_0) - f^*)}{2(k+1)} + \frac{\epsilon^2}{2} \geq \frac{1}{k+1}\sum_{i=0}^{k}\|\nabla f(\mathbf{x}_i)\|^2$$

Choose the smallest $k_0$ such that $\frac{\ell(f(\mathbf{x}_0)-f^*)}{(k_0+1)} \leq \epsilon^2$. Then we have

$$\epsilon^2 \geq \frac{1}{k_0+1}\sum_{i=0}^{k_0}\|\nabla f(\mathbf{x}_i)\|^2$$

Since the average of the squared norms of the gradients is less than $\epsilon^2$, there should be at least one that is less or equal to $\epsilon^2$. That is there is a $k \leq k_0$ such that $\|\nabla f(\mathbf{x}_k)\| \leq \epsilon$. Given the definition of $k_0$ we get the iteration bound stated in the theorem. $\qquad\square$

The last theorems give us a guarantee that the norm of the gradient is converging to zero but this is not enough to prove convergence to a single stationary point if $f$ has non isolated critical points. To establish a stronger result we prove that $\{\|\nabla f(\mathbf{x}_k)\|\}$ does not decrease arbitrarily quickly.

**Lemma 15** (Sufficiently large gradients)**.** *Suppose that $g_0$ is a $(L, B, c)$-well-behaved function for a $\ell$-gradient Lipschitz function $f$. Then we have that*

$$\|\nabla f(\mathbf{x}_{k+1})\| \geq (1 - \eta\ell)\|\nabla f(\mathbf{x}_k)\| - \eta\ell\|\varepsilon_k\|$$

*Proof.*

$$\begin{aligned}
\|\nabla f(\mathbf{x}_{k+1})\| &\geq \|\nabla f(\mathbf{x}_k)\| - \|\nabla f(\mathbf{x}_{k+1}) - \nabla f(\mathbf{x}_k)\| \\
&\geq \|\nabla f(\mathbf{x}_k)\| - \ell\|\mathbf{x}_{k+1} - \mathbf{x}_k\| \\
&\geq \|\nabla f(\mathbf{x}_k)\| - \eta\ell\|q_x(\mathbf{x}_k, h_k)\| \\
&\geq \|\nabla f(\mathbf{x}_k)\| - \eta\ell\|\nabla f(\mathbf{x}_k) + \varepsilon_k\| \\
&\geq \|\nabla f(\mathbf{x}_k)\| - \eta\ell\|\nabla f(\mathbf{x}_k)\| - \eta\ell\|\varepsilon_k\| \\
&\geq (1 - \eta\ell)\|\nabla f(\mathbf{x}_k)\| - \eta\ell\|\varepsilon_k\|
\end{aligned}$$

$\qquad\square$

Having established the above lemma we can use the Theorem 3.2 in [AMA05] and we are able to provide sufficient conditions to get convergence to a single stationary point even for functions with non isolated critical points.

**Theorem 9.** *Assume that $f$ is $\ell$-gradient Lipschitz, is analytic and that it has compact sub-level sets and that $g_0$ is a $(L, B, c)$-well-behaved gradient oracle. Let $\eta < \frac{1}{2\ell}$, $\beta < \frac{1-2\eta\ell}{B}$. Then $\lim \mathbf{x}_k$ exists and is a stationary point of $f$.*

*Proof.* We will first prove that given the fact that $f$ has compact sub-level sets $\{\mathbf{x}_k\}$ is confined in compact set. Based on Lemma 2 we have that for all $k \geq 0$

$$f(\mathbf{x}_{k+1}) - f(\mathbf{x}_k) \leq \frac{\eta}{2}\|\varepsilon_k\|^2$$

Applying this recursively and adding the inequalities

$$f(\mathbf{x}_{k+1}) \leq f(\mathbf{x}_0) + \frac{\eta}{2}\sum_{i=0}^{k}\|\varepsilon_i\|^2$$

$$\leq f(\mathbf{x}_0) + \frac{\eta}{2}\sum_{i=0}^{k}\left(c|h_0|(\beta B)^i\right)^2$$

$$\leq f(\mathbf{x}_0) + \frac{\eta}{2}c^2 h_0^2 \sum_{i=0}^{k}(\beta B)^{2i}$$

$$\leq f(\mathbf{x}_0) + \frac{\eta}{2}c^2 h_0^2 \frac{1}{1-(\beta B)^2}$$

So clearly $\{f(\mathbf{x}_k)\}$ is bounded and therefore $\{\mathbf{x}_k\}$ stays in one of the compact sub-level sets of $f$ forever.

Let us define the following

$$\phi_k(h_0) = c|h_0|(\beta B)^k$$

We will split the proof of the theorem in two cases. For the first case we will assume that there is a $k_0 \in \mathbb{N}$ such that

$$\|\nabla f(\mathbf{x}_{k_0})\| \geq \phi_{k_0}(h_0)$$

Then by Lemma 15

$$\|\nabla f(\mathbf{x}_{k_0+1})\| \geq (1-\eta\ell)\|\nabla f(\mathbf{x}_{k_0})\| - \eta\ell\|\varepsilon_{k_0}\|_2$$

$$\geq (1-\eta\ell)\phi_{k_0}(h_0) - \eta\ell\phi_{k_0}(h_0)$$

$$\geq (1-2\eta\ell)\phi_{k_0}(h_0)$$

$$\geq \frac{1-2\eta\ell}{\beta B}\beta B\phi_{k_0}(h_0)$$

$$\geq \frac{1-2\eta\ell}{\beta B}\phi_{k_0+1}(h_0)$$

$$\geq \phi_{k_0+1}(h_0)$$

By induction we have that $\forall k \geq k_0 + 1$

$$\|\nabla f(\mathbf{x}_k)\| \geq \frac{1-2\eta\ell}{\beta B}\phi_k(h_0)$$

By Lemma 14

$$\frac{\|\nabla f(\mathbf{x}_k)\|}{\|\varepsilon_k\|} \geq \left(\frac{1-2\eta\ell}{\beta B}\right) = q > 1$$

At the same time

$$-\nabla f(\mathbf{x}_k)^\top (\mathbf{x}_{k+1} - \mathbf{x}_k) = \eta \nabla f(\mathbf{x}_k)^\top (\nabla f(\mathbf{x}_k) + \varepsilon_k)$$
$$= \eta \|\nabla f(\mathbf{x}_k)\|^2 + \eta \nabla f(\mathbf{x}_k)^\top \varepsilon_k$$
$$\leq \eta \left(1 + \frac{1}{q}\right) \|\nabla f(\mathbf{x}_k)\|^2$$

Additionally using similar arguments as above

$$\frac{-\nabla f(\mathbf{x}_k)^\top (\mathbf{x}_{k+1} - \mathbf{x}_k)}{\|\nabla f(\mathbf{x}_k)\| \|(\mathbf{x}_{k+1} - \mathbf{x}_k)\|} \geq \frac{\eta \left(1 - \frac{1}{q}\right) \|\nabla f(\mathbf{x}_k)\|^2}{\eta \left(1 + \frac{1}{q}\right) \|\nabla f(\mathbf{x}_k)\|^2} = \frac{\left(1 - \frac{1}{q}\right)}{\left(1 + \frac{1}{q}\right)}$$

Let us define

$$c_1 = \frac{1}{2}\left(1 - \frac{1}{q}\right)$$

$$c_2 = \frac{\left(1 - \frac{1}{q}\right)}{\left(1 + \frac{1}{q}\right)}$$

Clearly by Lemma 2 we have that

$$f(\mathbf{x}_k) - f(\mathbf{x}_{k+1}) \geq \frac{\eta}{2}\left(\|\nabla f(\mathbf{x}_k)\|^2 - \|\varepsilon_k\|^2\right) \geq \frac{\eta}{2}\left(1 - \frac{1}{q^2}\right)\|\nabla f(\mathbf{x}_k)\|^2$$

We can conclude that

$$f(\mathbf{x}_k) - f(\mathbf{x}_{k+1}) \geq -c_1 \nabla f(\mathbf{x}_k)^\top (\mathbf{x}_{k+1} - \mathbf{x}_k) \geq c_1 c_2 \|\nabla f(\mathbf{x}_k)\| \|(\mathbf{x}_{k+1} - \mathbf{x}_k)\|$$

with $c_1 c_2 > 0$. Moreover, $\|\nabla f(\mathbf{x}_k)\| \geq \phi_k(h_0) > 0$ so we do not have to worry about arriving on stationary points in finite time. Given that $f$ is analytic, we have all the necessary conditions of Theorem 3.2 in [AMA05] and we have ruled out the possibility of $\{\mathbf{x}_k\}$ escaping to infinity. Therefore, we can now claim that $\{\mathbf{x}_k\}$ converges.

For the second case we have that for for all $k \in \mathbb{N}$

$$\|\nabla f(\mathbf{x}_k)\| < \phi_k(h_0).$$

We will now prove that $\{\mathbf{x}_k\}$ is a Cauchy sequence.

$$\|\mathbf{x}_k - \mathbf{x}_m\| \leq \sum_{i=m}^{k} \|\mathbf{x}_{i+1} - \mathbf{x}_i\|$$
$$\leq \sum_{i=m}^{k} \|\eta q_x(\mathbf{x}_i, h_i)\|$$
$$\leq \eta \sum_{i=m}^{k} \|\nabla f(\mathbf{x}_i, h_i) + \varepsilon_i\|$$
$$\leq 2\eta \sum_{i=m}^{k} \phi_i(h_0)$$

We know that $\sum_i^\infty \phi_i(h_0)$ converges so the partial sums must converge to 0. Then

$$\lim_{m,k \to \infty} \|\mathbf{x}_k - \mathbf{x}_m\| \leq 2\eta \lim_{m,k \to \infty} \sum_{i=m}^{k} \phi_i(h_0) = 0$$

So $\lim_{m,k \to \infty} \|\mathbf{x}_k - \mathbf{x}_m\| = 0$ and $\{\mathbf{x}_k\}$ is a Cauchy sequence bounded in a compact set and therefore it converges.

In either of the cases the limit of $\{\mathbf{x}_k\}$ is of course a stationary point. $\qquad \square$

We can now conclude our analysis with this final theorem.

**Theorem 10** (Theorem 3 restated). *Let $f : \mathbb{R}^d \to \mathbb{R} \in C^2$ be a $\ell$-gradient Lipschitz function. Let us also assume that $f$ is analytic, has compact sub-level sets and all of its saddle points are strict. Let $g_0$ be a $(L, B, c)$-well-behaved function for $f$ with $\eta < \min\{\frac{1}{L}, \frac{1}{2\ell}\}$ and $\beta < \frac{1-2\eta\ell}{B}$. If we pick a random initialization point $\mathbf{x}_0$, then we have that for the $\mathbf{x}_k$ iterates of $g_0$*

$$\forall h_0 \in \mathbb{R} \quad \Pr(\lim_{k \to \infty} \mathbf{x}_k = \mathbf{x}^*) = 1$$

*where $\mathbf{x}^*$ is a local minimizer of $f$.*

*Proof.* Given the assumptions, we can apply Theorem 9 and get that $\lim_{k \to \infty} \mathbf{x}_k$ exists and is a stationary point of $f$. We can also apply Theorem 1 in order to guarantee that the limit is not a strict saddle of $f$ with probability 1. Given the assumption that $f$ has only strict saddles, then $\lim_{k \to \infty} \mathbf{x}_k$ is with probability 1 a local minimum of $f$. $\qquad\square$

## C   Escaping Saddle Points Efficiently Detailed proofs

Before presenting the iteration complexity proof ( Theorem 4 ) we will state our main probabilistic lemma.

**Lemma 16.** *There exists an absolute constant $c_{\max}$, such that for any $f$ that is $\ell$-gradient Lipschitz and $\rho$-Hessian Lipschitz function and any $c \leq c_{\max}$, and $\chi \geq 1$. Let $\eta, r, g_{thres}, f_{thres}, t_{thres}, h_{low}$ be calculated same way as in Algorithm 1. Then, if $\mathbf{x}_t$ satisfies:*

$$\|\nabla f(\mathbf{x}_t)\| \leq g_{thres} \qquad and \qquad \lambda_{\min}(\nabla^2 f(\mathbf{x}_t)) \leq -\sqrt{\rho\epsilon}$$

*Let $\tilde{\mathbf{x}}_0 = \mathbf{x}_t + \boldsymbol{\xi}$, where $\boldsymbol{\xi}$ comes from the uniform distribution over $\mathcal{B}_0(r)$, and let $\{\tilde{\mathbf{x}}_i\}$ be the iterates of approximate gradient descent from $\tilde{\mathbf{x}}_0$ with stepsize $\eta$ and $h = h_{low}$, then with at least probability $1 - \frac{d\ell}{\sqrt{\rho\epsilon}} e^{-\chi}$, we have:*

$$\exists \quad i \leq t_{thres} : f(\mathbf{x}_t) - f(\tilde{\mathbf{x}}_{i'}) \geq f_{thres}$$

This lemma will be the "workhorse" which will offer the high probability guarantees of Algorithm 1 given that substantial progress can be made in the low gradient phase. The proof of the above lemma is deferred to the end of this section.

We are ready now to prove our main theorem:

**Theorem 11** (Theorem 4 restated). *There exists absolute constant $c_{\max}$ such that: if $f$ is $\ell$-gradient Lipschitz and $\rho$-Hessian Lipschitz, then for any $\delta > 0, \epsilon \leq \frac{\ell^2}{\rho}, \Delta_f \geq f(\mathbf{x}_0) - f^\star$, and constant $c \leq c_{\max}$, with probability $1 - \delta$, the output of $PAGD(x_0, \ell, \rho, \epsilon, c, \delta, \Delta_f)$ will be $\epsilon$-SOSP , and terminate in iterations:*

$$\mathcal{O}\left(\frac{\ell(f(\mathbf{x}_0) - f^\star)}{\epsilon^2} \log^4\left(\frac{d\ell\Delta_f}{\epsilon^2\delta}\right)\right)$$

*Proof.* Denote $\tilde{c}_{\max}$ to be the absolute constant allowed in Lemma 16. In this theorem, we let $c_{\max} = \min\{\tilde{c}_{\max}, 3/32\}$, and choose any constant $c \leq c_{\max}$.

In this proof, that Algorithm 1 returns a point $\mathbf{x}$ that satisfies the following condition:

$$\|\nabla f(\mathbf{x})\| \leq g_{\text{thres}} = \frac{\sqrt{c}}{\chi^2} \cdot \epsilon, \qquad \lambda_{\min}(\nabla^2 f(\mathbf{x})) \geq -\sqrt{\rho\epsilon} \tag{7}$$

Since $c \leq 1, \chi \geq 1$, we have $\frac{\sqrt{c}}{\chi^2} \leq 1$, which implies any $\mathbf{x}$ satisfies Equation (7) is also a $\epsilon$-SOSP .

Starting from $\mathbf{x}_0$, we know if $\mathbf{x}_0$ does not satisfy Equation 7, there are only two cases:

1. $\|\mathbf{z}_0\| = \left\|q\left(\mathbf{x}_0, \frac{g_{\text{thres}}}{4c_h}\right)\right\| > \frac{3}{4}g_{\text{thres}}$
   In this case, $\|\nabla f(\mathbf{x}_0)\| \geq \frac{g_{\text{thres}}}{2}$ and Algorithm 1 will not add perturbation. By Lemma 2:

   $$f(\mathbf{x}_0) - f(\mathbf{x}_1) \geq \frac{\eta}{2} \cdot (\|\nabla f(\mathbf{x}_0)\|^2 - \|\varepsilon_0\|^2)$$

   where $\varepsilon_0 = q\left(\mathbf{x}_0, \frac{g_{\text{thres}}}{4c_h}\right) - \nabla f(\mathbf{x}_0)$. Therefore we get $\|\varepsilon_0\| \leq \frac{g_{\text{thres}}}{4}$

   $$f(\mathbf{x}_0) - f(\mathbf{x}_1) \geq \frac{\eta}{2} \cdot (\|\nabla f(\mathbf{x}_0)\|^2 - \|\varepsilon_0\|^2) \geq \frac{3\eta}{32}g_{\text{thres}}^2 \geq \frac{3c^2\epsilon^2}{32\ell\chi^4}$$

2. $\|\mathbf{z}_0\| = \left\|q\left(\mathbf{x}_0, \frac{g_{\text{thres}}}{4c_h}\right)\right\| \leq \frac{3}{4}g_{\text{thres}}$
   In this case, $\|\nabla f(\mathbf{x}_0)\| \leq g_{\text{thres}}$ and Algorithm 1 will add a perturbation $\boldsymbol{\xi}$ of radius $r$ such that $\tilde{\mathbf{x}}_0 \leftarrow \mathbf{x}_0 + \boldsymbol{\xi}$, and will perform approximate gradient descent (without perturbations) for at most $t_{\text{thres}}$ steps. Since $\mathbf{x}_0$ is not a second-order stationary point then by Lemma 16 there exists $i' \leq t_{\text{thres}}$ such that:

   $$f(\mathbf{x}_0) - f(\mathbf{x}_1) = f(\mathbf{x}_0) - f(\tilde{\mathbf{x}}_{i'}) \geq f_{\text{thres}} = \frac{c}{\chi^3} \cdot \sqrt{\frac{\epsilon^3}{\rho}}$$

This means on average every step decreases the function value by

$$\frac{f(\mathbf{x}_0) - f(\tilde{\mathbf{x}}_{i'})}{i'} \geq \frac{f_{\text{thres}}}{t_{\text{thres}}} = \frac{c^3}{\chi^4} \cdot \frac{\epsilon^2}{\ell}$$

Hence, we can conclude that as long as Algorithm 1 has not terminated yet, on average, every step decreases function value by at least $\frac{c^3}{\chi^4} \cdot \frac{\epsilon^2}{\ell}$. However, we clearly can not decrease function value by more than $f(\mathbf{x}_0) - f^\star$, where $f^\star$ is the minimum value of $f$. This means Algorithm 1 must terminate within the following number of iterations:

$$\frac{f(\mathbf{x}_0) - f^\star}{\frac{c^3}{\chi^4} \cdot \frac{\epsilon^2}{\ell}} = \frac{\chi^4}{c^3} \cdot \frac{\ell(f(\mathbf{x}_0) - f^\star)}{\epsilon^2} = O\left(\frac{\ell(f(\mathbf{x}_0) - f^\star)}{\epsilon^2} \log^4\left(\frac{d\ell\Delta_f}{\epsilon^2\delta}\right)\right)$$

Finally, we have to ensure that the above statement holds with high probability. In the worst case scenario, in each outer-loop iteration the algorithm will be enforced to add a perturbation yielding a decrease of $f_{\text{thres}}$. Thus, the maximum number of perturbations are at most:

$$\frac{f(\mathbf{x}_0) - f^\star}{f_{\text{thres}}} = \frac{f(\mathbf{x}_0) - f^\star}{\frac{c}{\chi^3} \cdot \sqrt{\frac{\epsilon^3}{\rho}}}$$

Applying Lemma 16, we know that the guaranteed decrease of $f_{\text{thres}}$ happens with probability at least $1 - \frac{d\ell}{\sqrt{\rho\epsilon}}e^{-\chi}$ each time. By union bound, the probability that all perturbations satisfy the decrease guarantee is at least

$$1 - \frac{d\ell}{\sqrt{\rho\epsilon}}e^{-\chi} \cdot \frac{f(\mathbf{x}_0) - f^\star}{\frac{c}{\chi^3} \cdot \sqrt{\frac{\epsilon^3}{\rho}}} = 1 - \frac{\chi^3 e^{-\chi}}{c} \cdot \frac{d\ell(f(\mathbf{x}_0) - f^\star)}{\epsilon^2}$$

Recall our choice of $\chi = 3\max\{\log(\frac{d\ell\Delta_f}{c\epsilon^2\delta}), 4\}$. Since $\chi \geq 12$, we have $\chi^3 e^{-\chi} \leq e^{-\chi/3}$, this gives:

$$\frac{\chi^3 e^{-\chi}}{c} \cdot \frac{d\ell(f(\mathbf{x}_0) - f^\star)}{\epsilon^2} \leq e^{-\chi/3}\frac{d\ell(f(\mathbf{x}_0) - f^\star)}{c\epsilon^2} \leq \delta$$

which finishes the proof.

$\qquad\square$

What remains to be proven is why adding a perturbation is guaranteed to help the algorithm decrease the value of $f$ substantially with high probability. Following the proof strategy of [JGN$^+$17] we will define some additional notation. Let the condition number be the ratio of the Lipschitz constant of $\nabla f$ and the smallest negative eigenvalue of the Hessian of $\mathbf{x}_t$ before adding the perturbation, i.e $\kappa = \ell/\gamma \geq 1$. Additionally we define the following units:

$$p \leftarrow \log(\tfrac{d\kappa}{\delta}), \mathfrak{L} \leftarrow \eta\ell, \mathfrak{F} \leftarrow \frac{\mathfrak{L}}{p^3}\frac{\gamma^3}{\rho^2}, \mathfrak{G} \leftarrow \frac{\sqrt{\mathfrak{L}}}{p^2}\frac{\gamma^2}{\rho}, \mathcal{S} \leftarrow \frac{\sqrt{\mathfrak{L}}}{p}\frac{\gamma}{\rho}, \mathfrak{R} \leftarrow \frac{2\mathcal{S}}{\kappa p}, \mathfrak{T} \leftarrow \frac{p}{\eta\gamma}$$

Following the above definitions, it holds that: $\mathcal{S} = \sqrt{\frac{\mathfrak{F}p}{\gamma}} = \frac{\mathfrak{G}p}{\gamma}, \ell\mathfrak{R} = 2\mathfrak{G}$ and $\eta\mathfrak{T}\mathfrak{G} = \mathcal{S}$

(A): The first argument in this proof is that if the $\tilde{\mathbf{x}}_i$ iterates do not achieve a decrease of $2.5\mathfrak{F}$ in $\mathfrak{c}\mathfrak{T}$ steps then they must remain confined in a small ball around $\tilde{\mathbf{x}}_0$.

**Lemma 17.** *For any constant* $\mathfrak{c} \geq 3$*, define:*

$$T = \min\left\{ \inf_t \{t | f(\mathbf{u}_0) - f(\mathbf{u}_t) \geq 2.5\mathfrak{F}\}, \mathfrak{c}\mathfrak{T} \right\}$$

*then, for any* $\eta \leq 1/\ell$*, we have for all* $t < T$ *that* $\|\mathbf{u}_t - \mathbf{u}_0\| \leq 100(\mathcal{S} \cdot \mathfrak{c})$*.*

*Proof of Lemma 17.* Applying repeatedly Lemma 2, we get for $t < T$

$$f(\mathbf{u}_t) - f(\mathbf{u}_0) \leq -\frac{\eta}{2}\sum_{i=0}^{t}\left(\|\nabla f(\mathbf{u}_i)\|^2 - \|\varepsilon_i\|^2\right)$$

where

$$\varepsilon_i = q_x(\mathbf{u}_i, h_{low}) - \nabla f(\mathbf{u}_i).$$

By definition of $T$ we have that the function value of $f$ has not yet decreased by $2.5\mathfrak{F}$.

$$\frac{\eta}{2}\sum_{i=0}^{t}\|\nabla f(\mathbf{u}_i)\|^2 \leq f(\mathbf{u}_0) - f(\mathbf{u}_t) + \frac{\eta}{2}\sum_{i=0}^{t}\|\varepsilon_i\|^2$$

$$\frac{\eta}{2}\sum_{i=0}^{t}\|\nabla f(\mathbf{u}_i)\|^2 \leq 2.5\mathfrak{F} + \frac{\eta}{2}\sum_{i=0}^{t}\|\varepsilon_i\|^2$$

Since $T \leq \mathfrak{c}\mathfrak{T}$ and also $\|\varepsilon_i\| \leq \mathfrak{G}$ we then have

$$\frac{\eta}{2}\sum_{i=0}^{t}\|\nabla f(\mathbf{u}_i)\|^2 \leq 2.5\mathfrak{F} + \frac{\eta}{2}\mathfrak{G}^2\mathfrak{c}\mathfrak{T}$$

$$\sum_{i=0}^{t}\|\nabla f(\mathbf{u}_i)\|^2 \leq \frac{5}{\eta}\mathfrak{F} + \mathfrak{G}^2\mathfrak{c}\mathfrak{T}$$

$$\sum_{i=0}^{t}\left(\|\nabla f(\mathbf{u}_i)\|^2 + \|\varepsilon_i\|^2\right) \leq \frac{5}{\eta}\mathfrak{F} + 2\mathfrak{G}^2\mathfrak{c}\mathfrak{T}$$

We also have that $\|q_x(\mathbf{u}_i, h_{low})\|^2 \leq 2\left(\|\nabla f(\mathbf{u}_i)\|^2 + \|\varepsilon_i\|^2\right)$. Therefore we have that

$$\sum_{i=0}^{t}\|q_x(\mathbf{u}_i, h_{low})\|^2 \leq \frac{10}{\eta}\mathfrak{F} + 4\mathfrak{G}^2\mathfrak{c}\mathfrak{T}$$

Now we can bound the difference between $\mathbf{u}_t$ and $\mathbf{u}_0$:

$$\|\mathbf{u}_t - \mathbf{u}_0\|^2 = \left\|\sum_{i=1}^{t}\mathbf{u}_i - \mathbf{u}_{i-1}\right\|^2$$

$$\leq t\sum_{i=1}^{t}\|\mathbf{u}_i - \mathbf{u}_{i-1}\|^2$$

$$\leq t\eta^2\sum_{i=0}^{t}\|q_x(\mathbf{u}_i, h_{low})\|^2$$

$$\leq t\eta^2\left(\frac{10}{\eta}\mathfrak{F} + 4\mathfrak{G}^2\mathfrak{c}\mathfrak{T}\right)$$

$$\leq t\eta^2\left(\frac{10}{\eta}\mathfrak{F} + 4\mathfrak{G}^2\mathfrak{c}\mathfrak{T}\right)$$

$$\leq \mathfrak{c}\mathfrak{T}\eta^2\left(\frac{10}{\eta}\mathfrak{F} + 4\mathfrak{G}^2\mathfrak{c}\mathfrak{T}\right)$$

Manipulating the constants we get

$$\|\mathbf{u}_t - \mathbf{u}_0\|^2 \leq \left(10\mathfrak{c} + \mathfrak{c}^2\right)\mathcal{S}^2$$

$$\|\mathbf{u}_t - \mathbf{u}_0\| \leq \sqrt{(10\mathfrak{c} + \mathfrak{c}^2)}\mathcal{S}$$

For any $\mathfrak{c} \geq 3$ we have

$$\|\mathbf{u}_t - \mathbf{u}_0\| \leq 100(\mathfrak{c}\mathcal{S})$$

$\square$

(B):The second step in our proof strategy is to show that if all the iterates from $\mathbf{u}_0$ are constrained in a small ball, iterates from $\mathbf{w}_0 = \mathbf{u}_0 + \mu \cdot \frac{\mathfrak{R}}{2} \mathbf{e}_1$, for large enough $\mu$ must be able to decrease the function value. In order to do that, we keep track of vector $\mathbf{v}$ which is the difference between $\{\mathbf{u}_i\}$ and $\{\mathbf{w}_i\}$. We also decompose $\mathbf{v}$ into two different eigenspaces: the direction $\mathbf{e}_1$ (the minimum-eigenvalue eigenvector) and its orthogonal subspace.

**Lemma 18.** *There exists absolute constant $c_{\max}, \mathfrak{c}$ such that: for any $\delta \in (0, \frac{d\kappa}{e}]$, let $f(\cdot), \hat{\mathbf{x}}$ satisfies the following conditions*

$$\|\nabla f(\hat{\mathbf{x}})\| \leq \mathfrak{G} \qquad and \qquad \lambda_{\min}(\nabla^2 f(\hat{\mathbf{x}})) \leq -\gamma$$

*and any two sequences $\{\mathbf{u}_t\}, \{\mathbf{w}_t\}$ with initial points $\mathbf{u}_0, \mathbf{w}_0$ satisfying:*

$$\mathbf{w}_0 = \mathbf{u}_0 + \mu \cdot \frac{\mathfrak{R}}{2} \cdot \mathbf{e}_1, \quad \mu \in [\delta/(2\sqrt{d}), 1], \quad \|\mathbf{u}_0 - \hat{\mathbf{x}}\| \leq \frac{\mathfrak{R}}{2}$$

*$\mathbf{e}_1$ is the eignevector of the minimum eigenvalue of $\nabla^2 f(\hat{\mathbf{x}})$. Assume also that $h_{low} \leq \frac{\rho S \delta}{2 c_h \sqrt{d}} \frac{\mathfrak{R}}{2}$. Define*

$$T = \min \left\{ \inf_t \{t | f(\mathbf{w}_0) - f(\mathbf{w}_t) \geq 2.5\mathfrak{F}\}, \mathfrak{c}\mathfrak{T} \right\}$$

*then, for any $\eta \leq c_{\max}/\ell$, if $\|\mathbf{u}_t - \mathbf{u}_0\| \leq 100(S \cdot \mathfrak{c})$ for all $t < T$, we will have $T < \mathfrak{c}\mathfrak{T}$.*

*Proof of Lemma 18.* Recall notation $\tilde{H} = \nabla^2 f(\hat{\mathbf{x}})$. Since $\delta \in (0, \frac{d\kappa}{e}]$, we always have $p \geq 1$. Define $\mathbf{v}_t = \mathbf{w}_t - \mathbf{u}_t$, by assumption, we have $\mathbf{v}_0 = \mu\frac{\mathfrak{R}}{2}\mathbf{e}_1$. Let us firstly define the gradient approximation errors for these two sequences

$$\varepsilon_{\mathbf{w}_t} = q_x(\mathbf{w}_t, h_{low}) - \nabla f(\mathbf{w}_t)$$
$$\varepsilon_{\mathbf{u}_t} = q_x(\mathbf{u}_t, h_{low}) - \nabla f(\mathbf{u}_t)$$

Now, consider the update equation for $\mathbf{w}_t$:

$$
\begin{aligned}
\mathbf{u}_{t+1} + \mathbf{v}_{t+1} =& \mathbf{w}_{t+1} \\
=& \mathbf{w}_t - \eta q_x(\mathbf{w}_t, h_{low}) \\
=& \mathbf{w}_t - \eta(\nabla f(\mathbf{w}_t) + \varepsilon_{\mathbf{w}_t}) \\
=& \mathbf{u}_t + \mathbf{v}_t - \eta \nabla f(\mathbf{u}_t + \mathbf{v}_t) - \eta \varepsilon_{\mathbf{w}_t} \\
=& \mathbf{u}_t + \mathbf{v}_t - \eta \nabla f(\mathbf{u}_t) - \eta \left[ \int_0^1 \nabla^2 f(\mathbf{u}_t + \theta \mathbf{v}_t) \mathrm{d}\theta \right] \mathbf{v}_t - \eta \varepsilon_{\mathbf{w}_t} \\
=& \mathbf{u}_t + \mathbf{v}_t - \eta \nabla f(\mathbf{u}_t) - \eta(\tilde{H} + \Delta_t')\mathbf{v}_t - \eta \varepsilon_{\mathbf{w}_t} \\
=& \mathbf{u}_t - \eta \nabla f(\mathbf{u}_t) + (I - \eta \tilde{H} - \eta \Delta_t')\mathbf{v}_t - \eta \varepsilon_{\mathbf{w}_t} \\
=& \mathbf{u}_t - \eta(\nabla f(\mathbf{u}_t) + \varepsilon_{\mathbf{u}_t}) + (I - \eta \tilde{H} - \eta \Delta_t')\mathbf{v}_t - \eta(\varepsilon_{\mathbf{w}_t} - \varepsilon_{\mathbf{u}_t}) \\
=& \mathbf{u}_t - \eta q_x(\mathbf{u}_t, h_{low}) + (I - \eta \tilde{H} - \eta \Delta_t')\mathbf{v}_t - \eta(\varepsilon_{\mathbf{w}_t} - \varepsilon_{\mathbf{u}_t}) \\
=& \mathbf{u}_{t+1} + (I - \eta \tilde{H} - \eta \Delta_t')\mathbf{v}_t - \eta(\varepsilon_{\mathbf{w}_t} - \varepsilon_{\mathbf{u}_t})
\end{aligned}
$$

where

$$\Delta_t' = \int_0^1 \nabla^2 f(u_t + \theta v_t)\mathrm{d}\theta - \tilde{H}$$

This gives the dynamic for $\mathbf{v}_t$ satisfy:

$$\mathbf{v}_{t+1} = (I - \eta \tilde{H} - \eta \Delta_t')\mathbf{v}_t - \eta(\varepsilon_{\mathbf{w}_t} - \varepsilon_{\mathbf{u}_t}) \tag{8}$$

Since $f$ is Hessian Lipschitz, we have

$$\|\Delta_t'\| = \left\| \int_0^1 \nabla^2 f(\mathbf{u}_t + \theta \mathbf{v}_t) - \nabla^2 f(\hat{\mathbf{x}})\mathrm{d}\theta \right\| \leq \int_0^1 \rho \|\mathbf{u}_t + \theta \mathbf{v}_t - \hat{\mathbf{x}}\|\mathrm{d}\theta \leq \rho(\|\mathbf{u}_t - \mathbf{u}_0\| + \|\mathbf{v}_t\| + \|\hat{\mathbf{x}} - \mathbf{u}_0\|).$$

For $t < T$ the sequence $\{\mathbf{w}_t\}$ has not decreased the function $f$ by $-2.5\mathfrak{F}$. In other words, it holds that $f(\mathbf{w}_0) - f(\mathbf{w}_t) \leq 2.5\mathfrak{F}$, so applying Lemma 17, we know for all $t \leq T$

$$\|\mathbf{w}_t - \mathbf{w}_0\| \leq 100(\mathcal{S}\mathfrak{c}).$$

By condition of Lemma 18, we know $\|\mathbf{u}_t - \mathbf{u}_0\| \leq 100(\mathcal{S}\mathfrak{c})$ for all $t < T$. This gives for all $t < T$:

$$
\begin{aligned}
\|\mathbf{v}_t\| = \|\mathbf{w}_t - \mathbf{u}_t\| = \|(\mathbf{w}_t - \mathbf{w}_0) - (\mathbf{u}_t - \mathbf{u}_0) + (\mathbf{w}_0 - \mathbf{u}_0)\| \\
\leq \|(\mathbf{w}_t - \mathbf{w}_0)\| + \|\mathbf{u}_t - \mathbf{u}_0\| + \|\mathbf{w}_0 - \mathbf{u}_0\| \\
\leq 100(\mathcal{S}\mathfrak{c}) + 100(\mathcal{S}\mathfrak{c}) + \mu\frac{\mathfrak{R}}{2} \\
\leq 200(\mathcal{S}\mathfrak{c}) + \frac{\mathfrak{R}}{2} \\
\leq (200\mathfrak{c} + 1)\mathcal{S} \quad\quad\quad (9)
\end{aligned}
$$

where the last step holds because $\frac{\mathfrak{R}}{2} \leq \mathcal{S}$ This gives us for $t < T$:

$$\|\Delta'_t\| \leq \rho(\|\mathbf{u}_t - \mathbf{u}_0\| + \|\mathbf{v}_t\| + \|\hat{\mathbf{x}} - \mathbf{u}_0\|) \leq \rho(100\mathfrak{c}\mathcal{S} + (200\mathfrak{c}+1)\mathcal{S} + \frac{\mathfrak{R}}{2}) \leq \rho\mathcal{S}(300\mathfrak{c}+2)$$

Let $\psi_t$ be the norm of $\mathbf{v}_t$ projected onto $\mathbf{e}_1$ direction and the normal vector and $\varphi_t$ correspondingly be the norm of $\mathbf{v}_t$ projected onto remaining subspace. Let us define as $\lambda = \eta\rho\mathcal{S}(300\mathfrak{c}+2)$. Equation 8 gives us:

$$\psi_{t+1} = \left\| \prod_{\mathbf{e}_1}(I - \eta\tilde{H})\mathbf{v}_t - \eta\Delta'_t\mathbf{v}_t - \eta(\boldsymbol{\varepsilon}_{\mathbf{w}_t} - \boldsymbol{\varepsilon}_{\mathbf{u}_t}) \right\|$$

$$\varphi_{t+1} = \left\| \prod_{\mathbb{R}^d\backslash\{\mathbf{e}_1\}}(I - \eta\tilde{H})\mathbf{v}_t - \eta\Delta'_t\mathbf{v}_t - \eta(\boldsymbol{\varepsilon}_{\mathbf{w}_t} - \boldsymbol{\varepsilon}_{\mathbf{u}_t}) \right\|$$

Lower bound of $\psi_{t+1}$:

$$
\begin{aligned}
\psi_{t+1} &= \left\| \prod_{\mathbf{e}_1}[(I - \eta\tilde{H})\psi_t\mathbf{e}_1 - \eta\Delta'_t\mathbf{v}_t - \eta(\boldsymbol{\varepsilon}_{\mathbf{w}_t} - \boldsymbol{\varepsilon}_{\mathbf{u}_t})] \right\| \\
&\geq \|(I - \eta\tilde{H})\psi_t\mathbf{e}_1\| - \eta\|\prod_{\mathbf{e}_1}[\Delta'_t\mathbf{v}_t]\| - \eta\|\prod_{\mathbf{e}_1}[\boldsymbol{\varepsilon}_{\mathbf{w}_t} - \boldsymbol{\varepsilon}_{\mathbf{u}_t}]\| \\
&\geq (1 + \gamma\eta)\psi_t - \eta\|\Delta'_t\mathbf{v}_t\| - \eta\|\boldsymbol{\varepsilon}_{\mathbf{w}_t} - \boldsymbol{\varepsilon}_{\mathbf{u}_t}\| \\
&\geq (1 + \gamma\eta)\psi_t - \eta\|\Delta'_t\|\|\mathbf{v}_t\| - \eta\|\boldsymbol{\varepsilon}_{\mathbf{w}_t} - \boldsymbol{\varepsilon}_{\mathbf{u}_t}\| \\
&\geq (1 + \gamma\eta)\psi_t - \lambda\sqrt{\psi_t^2 + \varphi_t^2} - \eta\|\boldsymbol{\varepsilon}_{\mathbf{w}_t} - \boldsymbol{\varepsilon}_{\mathbf{u}_t}\|
\end{aligned}
$$

Upper bound of $\varphi_{t+1}$:

$$
\begin{aligned}
\varphi_{t+1} &= \| \prod_{\mathbb{R}^d\backslash\{\mathbf{e}_1\}}[(I - \eta\tilde{H})\mathbf{v}_t - \eta\Delta'_t\mathbf{v}_t - \eta(\boldsymbol{\varepsilon}_{\mathbf{w}_t} - \boldsymbol{\varepsilon}_{\mathbf{u}_t})]\| \\
&\leq \| \prod_{\mathbb{R}^d\backslash\{\mathbf{e}_1\}}[(I - \eta\tilde{H})\mathbf{v}_t]\| + \| \prod_{\mathbb{R}^d\backslash\{\mathbf{e}_1\}}[\eta\Delta'_t\mathbf{v}_t]\| + \eta\| \prod_{\mathbb{R}^d\backslash\{\mathbf{e}_1\}}[\boldsymbol{\varepsilon}_{\mathbf{w}_t} - \boldsymbol{\varepsilon}_{\mathbf{u}_t}]\| \\
&\leq \| \prod_{\mathbb{R}^d\backslash\{\mathbf{e}_1\}}[(I - \eta\tilde{H})\mathbf{v}_t]\| + \|\eta\Delta'_t\mathbf{v}_t\| + \eta\|\boldsymbol{\varepsilon}_{\mathbf{w}_t} - \boldsymbol{\varepsilon}_{\mathbf{u}_t}\| \\
&\leq (1 + \gamma\eta)\varphi_t + \lambda\sqrt{\psi_t^2 + \varphi_t^2} + \eta\|\boldsymbol{\varepsilon}_{\mathbf{w}_t} - \boldsymbol{\varepsilon}_{\mathbf{u}_t}\|
\end{aligned}
$$

Therefore we have

$$\psi_{t+1} \geq (1+\gamma\eta)\psi_t - \lambda\sqrt{\psi_t^2 + \varphi_t^2} - \eta\|\varepsilon_{\mathbf{w}_t} - \varepsilon_{\mathbf{u}_t}\|$$

$$\varphi_{t+1} \leq (1+\gamma\eta)\varphi_t + \lambda\sqrt{\psi_t^2 + \varphi_t^2} + \eta\|\varepsilon_{\mathbf{w}_t} - \varepsilon_{\mathbf{u}_t}\|$$

We will now prove via induction the following fact:

**Claim 1.** $\forall t < T \quad \varphi_t \leq 4\lambda t \cdot \psi_t$ and $\|\varepsilon_{\mathbf{w}_t}\| \leq \frac{\lambda}{2\eta}\|\mathbf{v}_t\|$ and $\|\varepsilon_{\mathbf{u}_t}\| \leq \frac{\lambda}{2\eta}\|\mathbf{v}_t\|$

*Proof.* Let us prove the base case of the induction:

- By hypothesis of Lemma 18, we know $\varphi_0 = 0$ so $\varphi_0 \leq 4\lambda 0 \cdot \psi_0$ holds trivially

- Based on the choice of $h_{low}$ we have that

$$\|\varepsilon_{\mathbf{w}_t}\| \leq \rho\mathcal{S}\frac{\delta}{2\sqrt{d}}\frac{\mathfrak{R}}{2} \leq \frac{\lambda}{2\eta}\psi_0 \leq \frac{\lambda}{2\eta}\|\mathbf{v}_0\|$$

$$\|\varepsilon_{\mathbf{u}_t}\| \leq \rho\mathcal{S}\frac{\delta}{2\sqrt{d}}\frac{\mathfrak{R}}{2} \leq \frac{\lambda}{2\eta}\psi_0 \leq \frac{\lambda}{2\eta}\|\mathbf{v}_0\|.$$

Thus the base case of induction holds. Assume Claim 1 is true for $\tau \leq t$. Now we can rewrite the inequalities based on the inductive hypothesis as follows:

$$\psi_{t+1} \geq (1+\gamma\eta)\psi_t - 2\lambda\sqrt{\psi_t^2 + \varphi_t^2}$$

$$\varphi_{t+1} \leq (1+\gamma\eta)\varphi_t + 2\lambda\sqrt{\psi_t^2 + \varphi_t^2}$$

For $t+1 \leq T$, we have:

$$\left\{\begin{array}{ccc} 4\lambda(t+1)\psi_{t+1} & \geq & 4\lambda(t+1)\left((1+\gamma\eta)\psi_t - 2\lambda\sqrt{\psi_t^2 + \varphi_t^2}\right) \\ \varphi_{t+1} & \leq & 4\lambda t(1+\gamma\eta)\psi_t + 2\lambda\sqrt{\psi_t^2 + \varphi_t^2} \end{array}\right\}$$

Thus it suffices to prove that:

$$4\lambda t(1+\gamma\eta)\psi_t + 2\lambda\sqrt{\psi_t^2 + \varphi_t^2} \leq 4\lambda(t+1)\left((1+\gamma\eta)\psi_t - 2\lambda\sqrt{\psi_t^2 + \varphi_t^2}\right)$$

$$(2 + 8\lambda(t+1))\sqrt{\psi_t^2 + \varphi_t^2} \leq 4(1+\gamma\eta)\psi_t.$$

By choosing $\sqrt{c_{\max}} \leq \frac{1}{300\mathfrak{c}+2}\min\{\frac{1}{2\sqrt{2}}, \frac{1/3}{8\mathfrak{c}}\}$, using the facts $\begin{cases} \eta\rho\mathcal{S}\mathfrak{T} = \sqrt{\eta\ell} \\ \eta \leq c_{\max}/\ell \end{cases}$, we have

$$8\lambda(t+1) \leq 8\lambda T \leq 8\eta\rho\mathcal{S}(300\mathfrak{c}+2)\mathfrak{c}\mathfrak{T} = 8\sqrt{\eta\ell}(300\mathfrak{c}+2)\mathfrak{c} \leq 1/3$$

This gives:

$$4(1+\gamma\eta)\psi_t \geq 4\psi_t \geq \frac{7}{3}\sqrt{2\psi_t^2} \geq (2 + 8\lambda(t+1))\sqrt{\psi_t^2 + \varphi_t^2}$$

which finishes the induction of the first part.

Now, using again the induction hypothesis, we know $\varphi_t \leq 4\lambda t \cdot \psi_t \leq \psi_t$, this gives:

$$\psi_{t+1} \geq (1+\gamma\eta)\psi_t - \sqrt{2}\lambda\psi_t \geq (1 + \frac{\gamma\eta}{2})\psi_t \tag{10}$$

where the last step follows from

$$\sqrt{2}\lambda = \sqrt{2}\eta\rho\mathcal{S}(300\mathfrak{c}+2) = \sqrt{2}\sqrt{\eta\ell}\frac{\gamma}{\rho p} \leq \sqrt{c_{\max}}(300\mathfrak{c}+2)\gamma\frac{\eta}{p} < \frac{\gamma\eta}{2}.$$

Equation 10 yields that $\psi_t$ is increasing sequence. Clearly

$$\|\varepsilon_{\mathbf{w}_{t+1}}\| \leq \frac{\lambda}{2\eta}\psi_0 \leq \frac{\lambda}{2\eta}\psi_{t+1} \leq \frac{\lambda}{2\eta}\|\mathbf{v}_{t+1}\|$$

$$\|\varepsilon_{\mathbf{u}_{t+1}}\| \leq \frac{\lambda}{2\eta}\psi_0 \leq \frac{\lambda}{2\eta}\psi_{t+1} \leq \frac{\lambda}{2\eta}\|\mathbf{v}_{t+1}\|$$

Thus we have completed the induction. □

Finally, combining Eq.(9) and (10) we have for all $t < T$:

$$(200\mathfrak{c}+1)\mathcal{S} \geq \|\mathbf{v}_t\| \geq \psi_t \geq (1+\frac{\gamma\eta}{2})^t\psi_0 = (1+\frac{\gamma\eta}{2})^t\mu\frac{\mathfrak{R}}{2} = (1+\frac{\gamma\eta}{2})^t\frac{\mathcal{S}}{\kappa}\frac{1}{p} = (1+\frac{\gamma\eta}{2})^t\frac{\delta}{2\sqrt{d}}\frac{\mathcal{S}}{\kappa}\frac{1}{p}$$

This implies:

$$T < \frac{\log(\frac{(200\mathfrak{c}+1)}{2\sqrt{d}}\frac{\kappa d}{\delta}\cdot p)}{\log(1+\frac{\gamma\eta}{2})} \leq \frac{\log((200\mathfrak{c}+1))+\log(\frac{\kappa d}{\delta})+\log p}{(\frac{\gamma\eta}{2})} \leq \frac{2\log(200\mathfrak{c}+1)}{\gamma\eta}+2\frac{\log(\frac{\kappa d}{\delta})}{\gamma\eta}+2\frac{p}{\gamma\eta}$$

The last inequality is due to the following facts

- $p = \log(\frac{\kappa d}{\delta}) \geq 1$ and $\forall x \geq 1 : \log x \leq x$.
- $\forall x \geq 0 : \log(1+x) \leq x$ thus $\log(1+\frac{\gamma\eta}{2}) \leq \frac{\gamma\eta}{2}$.
- $\mathfrak{T} = \frac{p}{\gamma\eta}$

Therefore, it holds that:

$$T < 2\log(200\mathfrak{c}+1)\frac{p}{\gamma\eta} + 4\mathfrak{T} \leq \mathfrak{T}(2\log(200\mathfrak{c}+1)+4)$$

By choosing constant $\mathfrak{c}$ to be large enough to satisfy $2\log(200\mathfrak{c}+1)+4 \leq \mathfrak{c}$, for example (i.e $\mathfrak{c} \geq 21$), we will have $T < \mathfrak{c}\mathfrak{T}$, which finishes the proof. $\qquad\square$

(C): Until now we have proved that firstly if approximate gradient descent from $\mathbf{u}_0$ does not decrease function value, then all the iterates must lie within a small ball around $\mathbf{u}_0$ (Lemma 17) and secondly starting an approximate descent from $\mathbf{w}_0$, which is $\mathbf{u}_0$ but displaced along $\mathbf{e}_1$ direction (negative eigenvalue's eigenvector for at least a certain distance), will decreases the function value if $\{\mathbf{u}_t\}$ is bounded. (Lemma 18).

The following lemma combines the above two lemmas:

**Lemma 19.** *There exists a universal constant $\hat{c}_{\max}$, for any $\delta \in (0, \frac{d\kappa}{e}]$, let $f(\cdot), \hat{\mathbf{x}}$ satisfies the following conditions*

$$\|\nabla f(\hat{\mathbf{x}})\| \leq \mathfrak{G} \qquad and \qquad \lambda_{\min}(\nabla^2 f(\hat{\mathbf{x}})) \leq -\gamma$$

*and $\mathbf{e}_1$ be the minimum eigenvector of $\nabla^2 f(\hat{\mathbf{x}})$. Consider two algorithm sequences $\{\mathbf{u}_t\}, \{\mathbf{w}_t\}$ with initial points $\mathbf{u}_0, \mathbf{w}_0$ satisfying:*

$$\|\mathbf{u}_0 - \hat{\mathbf{x}}\| \leq \tfrac{\mathfrak{R}}{2}, \quad \mathbf{w}_0 = \mathbf{u}_0 + \mu \cdot \tfrac{\mathfrak{R}}{2} \cdot \mathbf{e}_1, \quad \mu \in [\delta/(2\sqrt{d}), 1]$$

*Then, for any step size $\eta \leq \hat{c}_{\max}/\ell$, at least one of the following is true*

- *there exists $T_u \leq \frac{1}{\hat{c}_{\max}} \mathfrak{T}$ such that $f(\mathbf{u}_0) - f(\mathbf{u}_{T_u}) \geq 2.5\mathfrak{F}$*

- *there exists $T_w \leq \frac{1}{\hat{c}_{\max}} \mathfrak{T}$ such that $f(\mathbf{w}_0) - f(\mathbf{w}_{T_w}) \geq 2.5\mathfrak{F}$*

*Proof of Lemma 19.* Let $(c_{\max}^{(1)}, \mathfrak{c})$ be the absolute constant so that Lemma 18 holds. Choose

$$\hat{c}_{\max} = \min\{1, c_{\max}^{(1)}, \frac{1}{\mathfrak{c}}\}$$

Let $T^\star = \mathfrak{c}\mathfrak{T}$. Notice that by definition $T^\star \leq \frac{1}{\hat{c}_{\max}} \mathfrak{T}$. Finally , define:

$$T^\circ = \inf_t \{t | f(\mathbf{u}_0) - f(\mathbf{u}_t) \geq 2.5\mathfrak{F}\}$$

Let's consider following two cases:

**Case $T^\circ \leq T^\star$:** Clearly for this case we have for $T_u = T^\circ$ that

$$f(\mathbf{u}_0) - f(\mathbf{u}_{T_u}) \geq 2.5\mathfrak{F}$$

**Case $T^\circ > T^\star$:** In this case, by Lemma 17, we know $\|u_t - u_0\| \leq O(\mathcal{S})$ for all $t \leq T^\star$. Define

$$T^{\circ\circ} = \inf_t \{t | f(\mathbf{w}_0) - f(\mathbf{w}_t) \geq 2.5\mathfrak{F}\}$$

By Lemma 18, we immediately have $T^{\circ\circ} \leq T^\star = \mathfrak{c}\mathfrak{T}$. Clearly for this case we have for $T_u = T^{\circ\circ}$ we have that

$$f(\mathbf{w}_0) - f(\mathbf{w}_{T_w}) \geq 2.5\mathfrak{F}.$$

$\square$

Having expanded the basic lemmas ((A),(B),(C)) of [JGN$^+$17] for the zero order case, we are able to use the basic geometric upper bound of the stuck region. For the sake of completeness we state again the main lemma:

**Lemma 20.** *Let $f$ be a $\ell$-gradient Lipschitz and $\rho$-Hessian Lipschitz function. There exists universal constant $c_{\max}$, for any $\delta \in (0, \frac{d\kappa}{e}]$, suppose we start with point $\hat{\mathbf{x}}$ satisfying following conditions:*

$$\|\nabla f(\hat{\mathbf{x}})\| \leq \mathfrak{G} \qquad and \qquad \lambda_{\min}(\nabla^2 f(\hat{\mathbf{x}})) \leq -\gamma$$

*Let $\mathbf{x}_0 = \hat{\mathbf{x}} + \boldsymbol{\xi}$ where $\boldsymbol{\xi}$ come from the uniform distribution over ball with radius $r = \frac{R}{2}$, and let $\mathbf{x}_t$ be the iterates of approximate gradient descent from $\mathbf{x}_0$ and $T = \frac{\mathfrak{T}}{c_{\max}}$. Then, when step size $\eta \leq c_{\max}/\ell$, with at least probability $1 - \delta$, we have that:*

$$\exists t \leq T : f(\hat{\mathbf{x}}) - f(\mathbf{x}_t) \geq \mathfrak{F}$$

*Proof of Lemma 20.* By adding perturbation, in worst case we increase function value by:

$$f(\mathbf{x}_0) - f(\hat{\mathbf{x}}) \leq \nabla f(\hat{\mathbf{x}})^\top \boldsymbol{\xi} + \frac{\ell}{2}\|\boldsymbol{\xi}\|^2 \leq \frac{3\ell}{8}\mathfrak{R}^2 = \frac{3\ell}{8}\frac{4\mathcal{S}^2}{\kappa^2 p^2} = \frac{3\ell}{2}\frac{\frac{\mathfrak{F}p}{\gamma}}{\kappa^2 p^2} \leq \frac{3}{2}\mathfrak{F}\frac{1}{\kappa p} \leq \frac{3}{2}\mathfrak{F}$$

We know $\mathbf{x}_0$ come from the uniform distribution over $\mathcal{B}_{\hat{\mathbf{x}}}(r)$. Let $\mathcal{A} \subset \mathcal{B}_{\hat{\mathbf{x}}}(r)$ denote the set of bad starting points

$$\mathcal{A} = \{\mathbf{x} \in \mathcal{B}_{\hat{\mathbf{x}}}(r)| \quad \forall t \leq T : \quad f(\mathbf{x}_0) - f(\mathbf{x}_t) < 2.5\mathfrak{F}\}$$

otherwise if $\mathbf{x}_0 \in B_{\hat{\mathbf{x}}}(r) \setminus \mathcal{A}$, we have that

$$\exists t \leq T : f(\mathbf{x}_0) - f(\mathbf{x}_t) \geq 2.5\mathfrak{F}$$

By applying Lemma 18, we know for any $\mathbf{x}_0 \in \mathcal{A}$, it is guaranteed that

$$\mathbf{x}_0 \pm \mu r \mathbf{e}_1 \notin \mathcal{A} \text{ where } \mu \in [\frac{\delta}{2\sqrt{d}}, 1]$$

where $\mathbf{e}_1$ is the eigenvector of $\nabla^2 f(\hat{\mathbf{x}})$ with the smallest negative eigenvalue.

Let us denote $I_{\mathcal{A}}(\cdot)$ be the indicator function of being inside set $\mathcal{A}$. For a vector $\mathbf{x}$ let us define the following quantities

$$x_{\mathbf{e}_1} = \langle \mathbf{x}, \mathbf{e}_1 \rangle$$

$$\mathbf{x}_{\neg \mathbf{e}_1} = \prod_{\mathbb{R}^d \setminus \{\mathbf{e}_1\}} \mathbf{x}$$

Recall $\mathcal{B}^{(d)}(r)$ be $d$-dimensional ball with radius $r$. By calculus, this gives an upper bound on the volume of $\mathcal{A}$:

$$\text{Vol}(\mathcal{A}) = \int_{\mathcal{B}_{\hat{\mathbf{x}}}^{(d)}(r)} d\mathbf{x} \cdot I_{\mathcal{A}}(\mathbf{x})$$

$$= \int_{\mathcal{B}_{\hat{\mathbf{x}}}^{(d-1)}(r)} d\mathbf{x}_{\neg \mathbf{e}_1} \int_{\hat{\mathbf{x}}_{\mathbf{e}_1} - \sqrt{r^2 - \|\hat{\mathbf{x}}_{\neg \mathbf{e}_1} - \mathbf{x}_{\neg \mathbf{e}_1}\|^2}}^{\hat{\mathbf{x}}_{\mathbf{e}_1} + \sqrt{r^2 - \|\hat{\mathbf{x}}_{\neg \mathbf{e}_1} - \mathbf{x}_{\neg \mathbf{e}_1}\|^2}} dx_{\mathbf{e}_1} \cdot I_{\mathcal{A}}(\mathbf{x})$$

$$\leq \int_{\mathcal{B}_{\hat{\mathbf{x}}}^{(d-1)}(r)} d\mathbf{x}_{\neg \mathbf{e}_1} \cdot \left(2 \cdot \frac{\delta}{2\sqrt{d}}r\right) = \text{Vol}(\mathcal{B}_0^{(d-1)}(r)) \times \frac{\delta r}{\sqrt{d}}$$

Then, we immediately have the ratio:

$$\frac{\text{Vol}(\mathcal{A})}{\text{Vol}(\mathcal{B}_{\hat{\mathbf{x}}}^{(d)}(r))} \leq \frac{\frac{\delta r}{\sqrt{d}} \times \text{Vol}(\mathcal{B}_0^{(d-1)}(r))}{\text{Vol}(\mathcal{B}_0^{(d)}(r))} = \frac{\delta}{\sqrt{\pi d}}\frac{\Gamma(\frac{d}{2} + 1)}{\Gamma(\frac{d}{2} + \frac{1}{2})} \leq \frac{\delta}{\sqrt{\pi d}} \cdot \sqrt{\frac{d}{2} + \frac{1}{2}} \leq \delta$$

The second last inequality is by the property of Gamma function that $\frac{\Gamma(x+1)}{\Gamma(x+1/2)} < \sqrt{x + \frac{1}{2}}$ as long as $x \geq 0$. Therefore, with at least probability $1 - \delta$, $\mathbf{x}_0 \notin \mathcal{A}$. In this case, we have that there exists a $t \leq T$:

$$f(\hat{\mathbf{x}}) - f(\mathbf{x}_t) = f(\hat{\mathbf{x}}) - f(\mathbf{x}_0) + f(\mathbf{x}_0) - f(\mathbf{x}_t)$$
$$\leq 2.5\mathfrak{F} - 1.5\mathfrak{F} \geq \mathfrak{F}$$

which finishes the proof. $\qquad \square$

It is easy to check that our initial Lemma 16 can be derived by substituting $\eta = \frac{c}{\ell}, \gamma = \sqrt{\rho\epsilon}, \delta = \frac{d\ell}{\sqrt{\rho\epsilon}}e^{\chi}$ and simply applying the definitions of $\mathfrak{G}, \mathfrak{T}, \mathfrak{F}, g_{\text{thres}}, t_{\text{thres}}, f_{\text{thres}}$ into Lemma 20.