[Reviews · NeurIPS 2019]

Reviewer 1



Strengths: A well-written paper, with natural flow of reasoning. First, problems with black-box approaches are pointed out. Then it is shown that if the algorithm for well-behaved functions converges, then it converges to the local minimum a.s. Then it’s shown that the algorithm converges a.s. Appendix: boxes with the current goal really help to follow the paper. Weaknesses: -- Experimental section seems redundant. It doesn’t contain precise data which one can use for comparison with other approaches. Convergence to a local minimum is proved, there is no need to show this experimentally. -- Why is parameter \beta needed? It is always present together with either B or g, and therefore can be removed. -- Section B.2 of the appendix is devoted to the proof that a choice of h_0 can be arbitrary, not necessarily random. It would be helpful to provide a motivation (why is this fact useful), since from practical point of view it’s not a problem to initialize h randomly. -- Line 260: “In a typical zero order approach, one could resort to expensive line searches to determine the appropriate value of the gradient approximation accuracy in each step which guarantees the decrease of f”. A reference or other clarification should be given. Suggestions: -- Briefly describe black-box approaches (FPSG, ZPSG) in appendix, before proving Lemmas 6 and 7. -- For proofs and theorems in appendix, which are similar to [JGN+17], it may be potentially better to specify this similarity (e.g. “Similarly to [JGN+17] we establish the following result”) -- Algorithm 1: why special symbol \bot is needed? It can be replaced with x_t.

Reviewer 2



The paper proposes a zero-order optimization method that avoids saddle points efficiently. This is a practically important problem since many of the practical problems are non-differentiable of it is too costly to compute the gradient. Previously published methods of this category are not efficient and scale as O(d^4) with respect to the number of dimensions. The proposed algorithm is is a medley of different finite difference methods combined to maintain converance and efficiency, I really liked the related work session that covers many of the most resent works on the second-order defivative and dereivative free optimization. Generally, overall the paper flows very well and it a pleasure to read. The problematic part for me is virtual lack of the experimental secion. The only real experiment is done in the two dimensional Rastrigin function, which is clear toy example. More importantly, this function is differentiable, which defeats the purpose of zero-order methods. In fact the paper only very briefly mentions the application of the zero-order methods (line 33-41). It is suprizing that none of these applications made it as a experiments for the proposed very practical algorithm.

Reviewer 3



This paper provides interesting results for zero-order methods in non-convex optimization. Under some common assumptions, it shows that the Approximate Gradient Descent algorithm leads us to a second-order stationary point asymptotically with probability 1. It also shows that the PAGD algorithm in this paper can speed up the convergence by escaping strict saddle points efficiently. The convergence guarantee matches that of first-order methods, and experiments were done to verify their theoretical findings. The results in this paper are improvements to previous results, e.g., [JGN+17], [LPP+17] and [JLGJ18]. Specifically, this paper extends previous results to zero-order methods where one do not have access to the gradient. The results are interesting because the zero-order case is harder than the previous ones. This paper is generally written in a clear way and it's easy to understand. However, the writing style of the proof sketch in part 4 and part 5 seems a bit different. In part 4, the authors seem to use a lot of theorems and lemmas, but in lemma 5 there are mostly words. I would suggest the authors hide some details of section 4. Typo: line 270, "i.e. that is" -> "i.e., ".

[Author Response · NeurIPS 2019]

We would first like to thank the reviewers for their insightful comments on our work.

Reviewer #1 and #3 suggested that our current experimental section was offering little added value on top of our theoretical analysis. Although our main motivations are theoretical (i.e. explain theoretically why zero-order algorithms can provide the same guarantees as the first-order counterparts in non-convex settings), we provide a new experimental section that we believe couples well both with our stated goals as well with the prior related work in the area. Given the reported empirical success of zero-order algorithms in many applications, the scope of our experiments was not to verify the success of AGD or PAGD in general practical settings but to demonstrate the effect of saddle points for the case of zero order algorithms (versus first order methods). To offer an even better illustration of the above point, we propose to replace the 2-d rastrigin function with a high dimensional version of octopus function as presented in [DLJ+17]. This function is particularly relevant to our setting as it possesses a sequence of saddle points. The authors of [DLJ+17] proved that gradient descent needs exponential time to avoid saddle points before converging to a local minimum. In contrast the perturbed version of gradient descent does not suffer from the same limitation. Figure 1 clearly shows that the zero-order versions have the same iteration performance with the first-order ones. In fact, AGD is shown to behave even better than GD in this example thanks to the noise induced by the gradient approximation. Thus our theoretical findings are verified even in this well-established and challenging benchmark.

Reviewer #1 asked us about the motivation behind our choice to study the case of an arbitrary $h_0$ selection in contrast to a random one. Our main motivation was to offer similar guarantees with the first order counterparts [LPP+17] , i.e the avoidance of saddle point stems from their instability and not from an extra random dimension. Additionally, in our experience practitioners of zero-order methods tend to use some fixed values based on the machine precision and not generally some randomly sampled numbers. Thus, providing these stronger guarantees, our result reflects better what actually happens in practice.

Reviewer #1 asked for a clarification about the cost of a line search alternative. The main intuition is that one can try progressively smaller values of $h$ until the value of $f$ is decreased [Torczon, V.J. (1997)]. Using Lemma 3 and property iii) of Definition 4, one can see that the required number of trials actually depends on the norm of the gradient at the current iterate. For convergence to first-order stationary points ,i.e ($\|\nabla f(\mathbf{x}_k)\| \leq \epsilon$), this is hardly a problem, since $\|f(x)\| \geq \epsilon$ until termination and finding an appropriate $h$ is trivial. However, for second-order stationary points, such trivial termination condition does not hold. Therefore $\|\nabla f(x)\|$ can be arbitrarily small and thus the sample complexity of this process may be unbounded in terms of $\frac{1}{\epsilon}$. One of the surprising contribution of our work is that there exist zero-order methods (like PAGD) that can escape saddle points in a bounded number of iterations with a fixed $h$.

Finally, all the minor typos spotted by the reviewers will be corrected for the camera ready version. Regarding the issues of presentation for both the main text and the appendix, we will be happy to follow the suggestions of Reviewer #1. Additionally, about the difference of the writing style between section 4 and 5, that Reviewer #4 mentioned, we plan to hide the details of Corollary 1, Lemma 2 and Theorem 3 expressing their main intuition in text.

Figure 1: Octopus function of $d = 15$. Parameters of the function $\tau = e$, $L = e$, $\gamma = 1$. Parameters of first order methods taken from [DLJ+17]. Zero order methods use symmetric differencing with $h = 0.01$

[Torczon, V.J. (1997)]. "On the convergence of pattern search algorithms" (PDF). SIAM Journal on Optimization.

[Meta-Review · NeurIPS 2019]

This paper gives a new analysis for 0th order optimization and show that they can avoid saddle points efficiently. The new algorithm is much faster than previous algorithms based on Gaussian smoothing. Comparing number of iterations instead of oracle calls is still slightly misleading.